# Primate retina trades single-photon detection for high-fidelity contrast encoding

Markku Kilpeläinen [1,2], Johan Westö[3], Jussi Tiihonen [2,3], Anton Laihi[2], Daisuke Takeshita [2], Fred Rieke[4] & Petri Ala-Laurila [2,3] ✉

How the spike output of the retina enables human visual perception is not fully understood. Here, we address this at the sensitivity limit of vision by correlating human visual perception with the spike outputs of primate ON and OFF parasol (magnocellular) retinal ganglion cells in tightly matching stimulus conditions. We show that human vision at its ultimate sensitivity limit depends on the spike output of the ON but not the OFF retinal pathway. Consequently, nonlinear signal processing in the retinal ON pathway precludes perceptual detection of single photons in darkness but enables quantal-resolution discrimination of differences in light intensity.

Vision at absolute threshold provides a remarkable example of how sensory performance can approach the fundamental limits of physics. Classical psychophysical experiments showed that dark-adapted humans can detect less than a dozen light quanta[1–3]. Despite decades of work, the neural mechanisms that define this remarkable performance remain poorly understood. A key unresolved question is whether there is a neural threshold (nonlinearity) somewhere in the visual pathway that eliminates noise but requires that a minimum number of quanta are absorbed for a stimulus to be detected perceptually. The current theory for human visual perception ("classical model", see Fig. 1a) proposes that signals initiated by each photon absorption propagate linearly or near-linearly through visual circuits in the retina before encountering a perceptual decision criterion or threshold in the brain. In this model, perception, with an appropriate criterion, can access signals produced by each individual photon[4,5]. Evidence from past psychophysics experiments[6–9] for this theory, however, is circumstantial and does not account for recent mechanistic findings on retinal circuit function at the lowest light levels[10–12].

Signals originating from single-photon absorptions traverse the mammalian retina through the conserved rod bipolar pathway across light levels ranging from absolute threshold to background lights that produce a few photon absorptions per rod per second[7,13–15]. In this pathway, rod signals are transmitted to ON and OFF retinal ganglion cells (RGCs) via rod bipolar cells and AII amacrine cells. A key feature of

the rod bipolar pathway is the increasing rod convergence from its input to its outputs[16]. This convergence means that photon absorptions in a small fraction of the rods can noticeably modulate retinal outputs, but only if noise added in retinal circuits is avoided. Indeed, recent work shows that signals traversing the ON (but not the OFF) branch of the rod bipolar pathway in both primate and mouse are subject to a thresholding nonlinearity that discards noise and many single-photon responses while transmitting signals originating from coincidence of two or more photons in a pool of ~1000 rods[10,11]. Selective manipulation of the sensitivity of ON pathways in transgenic mice shows that behavioral detection of dim lights at the sensitivity limit relies on the responses of ON RGCs even when OFF RGC responses would allow higher sensitivity[11].

These recent findings challenge the classical model for perception at the sensitivity limit. They suggest instead a new model in which perception relies on retinal outputs originating in the ON pathway, and those outputs are shaped by a thresholding nonlinearity that limits access to individual single-photon responses (see Fig. 1b). However, linking human visual perception to retinal ON and OFF pathway function requires a different approach than in mice, where genetic manipulations could be used to manipulate ON and OFF pathway sensitivities. Instead, we here test the classical model vs. the new model by comparing detection sensitivity – the ability to detect dim light flashes – with discrimination sensitivity – the ability to identify which of two flashes is brighter. Comparison of retinal and psychophysical performance in these two tasks in a two-alternative forced

[1]Department of Psychology and Logopedics, University of Helsinki, Helsinki, Finland. [2]Molecular and Integrative Biosciences Research Programme, University of Helsinki, Helsinki, Finland. [3]Department of Neuroscience and Biomedical Engineering, Aalto University, Espoo, Finland. [4]Department of Physiology and Biophysics, University of Washington, Seattle, WA, US. ✉e-mail: petri.ala-laurila@helsinki.fi

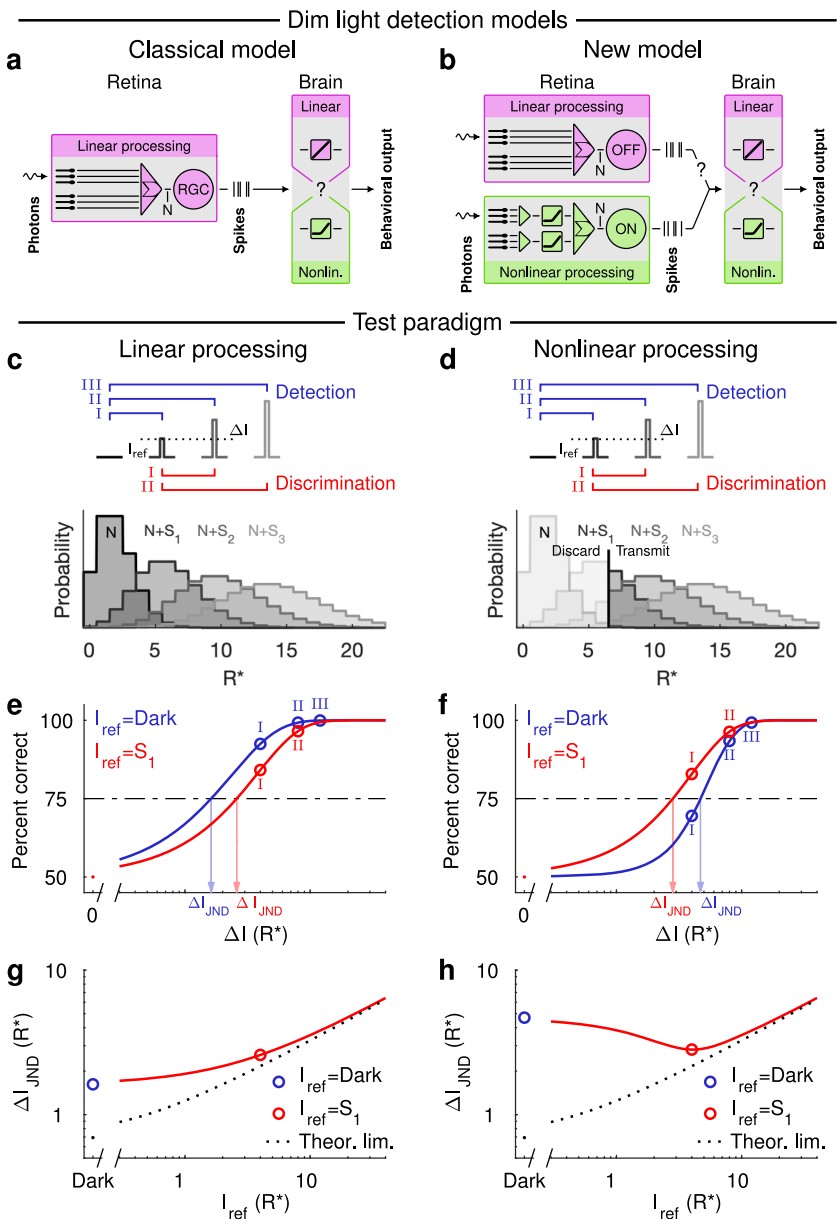

**Fig. 1 | Linear and nonlinear dim light detection models exhibit fundamentally different performance characteristics for detection and discrimination tasks. a** The classical dim light detection model assumes linear retinal processing and that behavioral sensitivity is only limited by noise and potentially a downstream thresholding mechanism beyond the retina. The post-retinal nonlinearity is adjustable allowing a tradeoff between sensitivity and false-positive rate. With the most lenient criterion (=highest false-positive rate), this nonlinearity disappears and perception has access to each absorbed photon[4]. **b** The new model, in turn, postulates that behavioral sensitivity is fundamentally limited by a thresholding mechanism along the retinal ON pathway and that the linear OFF pathway does not contribute to this detection task, but serves other visual functions (denoted by question mark). **c, d** The presence of thresholding nonlinearities can be observed by evaluating the dimmest light increment needed for detection (blue) and discrimination (red) of dim lights in a two-alternative forced choice (2AFC) setting.

Linear processing (**c**) allows an ideal observer to access all isomerization events ($R^*$; Poisson distributed signals): $S_1$ (mean = 4 $R^*$), $S_2$ (mean = 8 $R^*$) and $S_3$ (mean = 12 $R^*$) and noise $N$ (mean = 2 $R^*$), whereas a threshold (6 $R^*$) (**d**) restricts access to the events that surpass the threshold. **e, f** Ideal 2AFC performance for detection (blue) of a dim flash against a background noise (N), and for discriminating (red) a probe flash from a reference flash ($S_1$). For linear processing (**e**), detection is easier than discrimination (quantified by the just noticeable difference; $\Delta I_{JND}$), whereas the situation is reversed for nonlinear processing (**f**; the blue curve is now on the right side of the red one). **g, h** The $\Delta I_{JND}$ as a function of the reference flash intensity: the detection task (blue dot) corresponds to blank reference flash. The dotted black line shows the theoretical performance limit set by quantum fluctuations in the number of stimulus photons. The dip in **h** occurs approximately at a reference intensity of 4 $R^*$, as 4 additional $R^*$s are needed to surpass a threshold of 6 $R^*$ when the noise level is 2 $R^*$. Source data are provided as a Source Data file.

choice setting (2AFC) supports the new model in which the brain has access only to signals resulting from absorption of multiple photons. Further, our results suggest that behavioral detection of light increments in humans relies on the retinal ON pathway, which enables discrimination performance approaching the fundamental limits of physics.

## Results

### The "classical" and the "new" model perform differently on the discrimination task

Figure 1c–h illustrate predicted 2AFC task performance for linear and nonlinear models. Figure 1c shows the response (signal + noise) distributions for three flash strengths assuming linear processing.

Figure 1d applies a threshold to these distributions. For a linear system, detection of a flash in darkness (blue trace, Fig. 1e) is always easier than discrimination between two flashes with a corresponding difference in flash strength (red trace, Fig. 1e). This is because Poisson distributions arising from the photon statistics of light get wider as the flash strength increases, causing their overlap to increase for a fixed difference in flash strength (Fig. 1c). Elimination of small responses by nonlinear processing, however, can lower sensitivity in the detection task without affecting the ability to discriminate between suprathreshold responses. This can permit discrimination of a difference in strength between two dim flashes that is smaller than the detection threshold (Fig. 1f). This phenomenon creates a dip when the just noticeable intensity difference ($\Delta I_{JND}$), i.e., the increment in flash strength ($I_{probe} - I_{ref}$) corresponding to 75% correct, is plotted as a function of the intensity of the reference flash (Fig. 1h). Such dips have been reported in high-light-level contrast discrimination tasks for retinal ganglion cells[17], and their mechanistic origin could be understood in the context of nonlinear subunits in their receptive fields[18–22]. Similarly, dips have been previously reported for human psychophysics in various stimulus conditions[23–25]. This dip does not occur for a linear system with additive or multiplicative noise, where the function instead exhibits a monotonic increase (see Fig. 1g and Supplementary Fig. 3). The dip therefore is an identifying feature of a nonlinear system.

### ON parasols perform better than OFF parasols on the discrimination task

We start by describing the detection and discrimination performance of primate ON and OFF parasol (magnocellular-projecting) RGCs. ON and OFF parasol RGCs are likely of direct relevance for human psychophysics at its sensitivity limit due to their high contrast sensitivity and abundant rod input (see Ala-Laurila and Rieke[10]). Primate ON and OFF parasol RGCs are also the closest functional[15] and genetic[26] homologs of ON and OFF alpha RGCs in mice: primate parasols being the closest genetic homologs of the transient alpha RGCs in mice and primate midget RGCs being the closest genetic homologs of the sustained alpha RGCs in mice. ON and OFF alpha RGCs are also the most sensitive RGCs among all types in the mouse retina[11]. ON and OFF parasols are thereby good proxies among primate RGC types for highly sensitive readouts of the rod bipolar pathway[10,15].

We recorded spiking activity (cell-attached) of dark-adapted ON and OFF parasol RGCs (Fig. 2a inset) in response to sequences of dim flashes (Fig. 2a, b). We then used an ideal observer analysis to quantify detection (Fig. 2c, d; blue symbols) and discrimination performance (Fig. 2c, d, red symbols). As in Fig. 1, we defined performance as the light intensity difference ($\Delta I_{JND}$) needed to distinguish the brighter probe flash from the dimmer reference flash ($I_{ref}$, which was 0, i.e. darkness, in the detection task) in 75% of the trials.

ON and OFF parasol RGC performance differed fundamentally. OFF parasol RGCs had better detection performance than discrimination performance, consistent with linear processing (Fig. 2c). ON parasol RGCs, however, had better discrimination performance than detection performance (Fig. 2d), consistent with nonlinear processing. Further, the $\Delta I_{JND}$ as a function of the reference flash intensity rose monotonically for OFF parasol responses and exhibited a clear dip for ON parasol responses (Fig. 2e, f). The ON parasol performance approached the theoretical limit set by Poisson fluctuations of flash-induced visual pigment activations in the discrimination task but not the detection task (dashed black lines in Fig. 2e, f). OFF parasol performance was worse than that of ON parasol cells for discrimination ($P = 0.0007$, Welch's $t$-test, Cohen's $d = 2.55$) but similar for detection ($P = 0.15$, Welch's $t$-test; Cohen's $d = 0.41$; blue symbols in Fig. 2e, f, see also Ala-Laurila and Rieke[10]).

### Human perception follows the nonlinear ON parasol output with a post-retinal nonlinearity

How do the differences in detection vs discrimination found in responses of ON and OFF parasol RGCs relate to perception? To answer this question, we evaluated the psychophysical performance of dark-adapted human observers using the same 2AFC tasks. In the detection task, the observer reported whether a dim flash was present in the first or the second of two stimulus intervals (see Detection in Fig. 3a). In the discrimination task, the observer reported which of the two intervals contained a brighter flash (see Discrimination in Fig. 3a). Discrimination performance exceeded detection performance (Fig. 3b, $P < 0.01$, Student's $t$-test, Cohen's $d > 2$, for all observers, see "Methods" section), leading to a clear dip in the $\Delta I_{JND}$ function (Fig. 3c). This was the case for all five observers, and the average discrimination performance came close to the theoretical limit (black dashed line in Fig. 3c), while the detection performance was far from it. Human performance on the discrimination task was thus superior to that of OFF parasol RGCs but slightly inferior to that of ON parasol RGCs (Fig. 3d), supporting the dependence of behavior on responses of ON rather than OFF parasol cells under these conditions and in line with earlier findings in mice[11].

Although both human behavior and ON parasol RGCs exhibit nonlinear processing, the dip in the $\Delta I_{JND}$ function for human behavior is located at higher intensities than that for ON parasol RGCs. This change in the location of the dip suggests additional thresholding beyond the retina. We tested this hypothesis by comparing two distinct models: one with a post-retinal nonlinearity in the brain (Fig. 3e, M2) and one without it (M1). These models were based on ON RGCs because their discrimination performance was better than that of OFF RGCs. The aim was to test whether a simple model could link human psychophysics at the sensitivity limit of vision to retinal RGC outputs. The retinal components of both models are identical, with single-photon signals and noise (additive and multiplicative) being summed and thresholded by retinal subunits corresponding to ON-cone bipolar cells, as in Ala-Laurila and Rieke[10]. The retinal nonlinearity is followed by a second nonlinearity in M2, whereas M1 has a linear readout of the retina. For both models, we first fitted (least squares) the number of subunits ($n$), the additive and multiplicative noise ($N$), and the threshold ($\theta_1$) to describe the average performance of the ON parasols (Fig. 3e; M1), leading to $n = 3$ and $\theta_1 = 2$, close to earlier estimates[10]. This shows that a coincidence of a minimum of two single-photon responses in a subunit of ~2000 rods is required to create the response properties of ON parasol ganglion cells (see "Methods" section). Next, we tested if access to more than a single RGC could explain the difference between behavior and ON parasol responses, since the retinal stimulus size in psychophysics corresponds to ~2 times the RGC receptive field. Increasing the number of subunits, as demonstrated by doubling their number from 3 to 6 (M1, $n = 6$, see Fig. 3e) only shifted the curve upward in M1 but did not increase the dip. However, adding the second nonlinearity as in M2, resulted in a good fit to the human behavioral data (Fig. 3e; M2, $n = 6$, $\theta_1 = 2$, $\theta_2 = 2$; for model robustness, see Supplementary Fig. 4). This result is consistent with additional downstream thresholding of ON parasol signals in the brain at the sensitivity limit of vision.

## Discussion

Our results provide three key insights related to the mechanistic and functional underpinnings of human vision at the lowest light levels. First, similar to previous results in mice[11], our results support the hypothesis that human behavior at absolute threshold relies on retinal output signals provided by ON RGCs. This is true even when the sensitivity of OFF RGC responses is comparable to that of ON RGC responses. This suggests that, at least at visual threshold, increases in

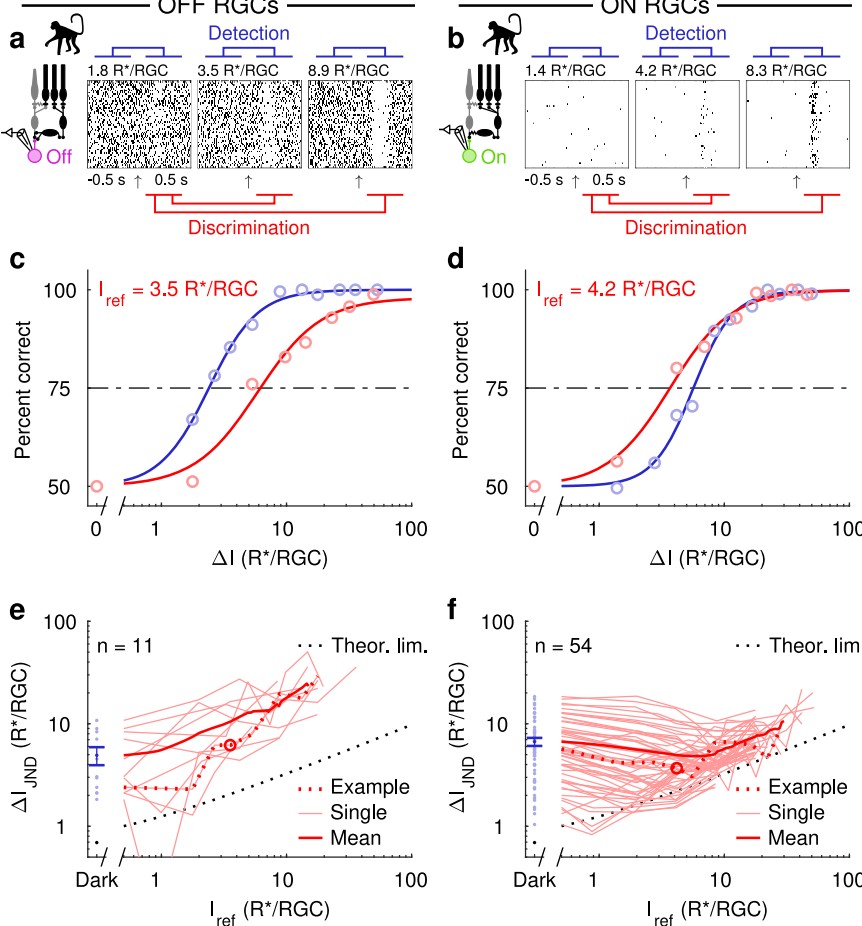

**Fig. 2 | The retinal OFF pathway is linear whereas the ON pathway is nonlinear.**
**a**, **b** Responses of example OFF (**a** magenta) and ON (**b** green) parasol RGCs as the retinal rod bipolar circuit (black) outputs to a brief dim flash (arrow indicates flash onset). An ideal observer performs the detection or the discrimination tasks by comparing the recorded spike responses. In the detection task, spontaneous activity is compared to flash responses, whereas responses to a reference flash are compared to responses to brighter flashes in the discrimination task.
**c**, **d** Performance of the OFF (**c**) and ON (**d**) RGCs on the detection task (blue) and on the discrimination task for one reference intensity (red). Markers indicate values computed from measured responses (see "Methods" section), and the continuous

lines are fitted Hill functions. The $\Delta I_{JND}$ is the increase in flash intensity ($\Delta I$) required to reach 75% correct (dashed line). **e**, **f** $\Delta I_{JND}$ as a function of the reference flash intensity (0 for detection) for all OFF (**e**) and ON (**f**) RGCs. The thin red lines correspond to individual cells, the dotted red lines to the example cells, the thick red lines to population averages, and the dotted black line shows the theoretical performance limit set by quantum fluctuations in the number of stimulus photons. Light blue circles correspond to individual cell performance in the detection task, and dark blue circles show the mean ± SEM (R*/RGC): 4.9 ± 1.0 (OFF parasols, $n = 11$ cells); 6.7 ± 0.6 (ON parasols, $n = 54$ cells). Source data are provided as a Source Data file.

retinal firing rates (e.g., ON RGC responses to light increments) are more effective than decreases (OFF RGC responses to increments) in eliciting cortical responses and behavior. Indeed, earlier results on primates at photopic light levels suggest that cortical responses to light increments originate in ON RGCs and those to light decrements originate in OFF RGCs[27]. This distribution of labor between the retinal ON and OFF pathways in driving cortical responses will be important to consider in attempts to restore visual signals in visually impaired patients. However, it still remains to be seen if behavioral detection of the weakest light decrements in humans relies on the retinal OFF pathway, as our recent results on mice suggest[12]. Such a study on humans will have to consider the fundamental limit set by the intensity of background light for generating quantal shadow stimuli (the dimmest light decrements), as shown previously[12].

Second, our results answer the decades-old question about whether humans can perceive a single photon. A recent study[5] using a single-photon light source reported that humans can detect even a single photon. Two issues make this conclusion uncertain: (1) The probability of detection was very low, and only marginally statistically significant, (2) the paper does not provide a model that accounts for

performance in both the single-photon detection task and the classical psychophysics task, which spans flash strengths in which detection performance ranges from chance to the near-perfect level. We have repeated near-identical experiments in an ongoing study (Tiihonen et al. in preparation) using both single-photon and conventional light sources; we used more trials and more near-threshold flash strengths than the previous study so that we could resolve small differences from chance performance. Human performance in these experiments did not exceed chance levels in the single-photon detection task and the full set of results were well described by the new model presented here (Fig. 1b).

Our results here indicate that responses to individual photons in the retina sum nonlinearly, providing evidence for the perceptual relevance of detecting coincident photons[3]. Thus, the biological circuit design appears to be optimized for extraordinary discrimination of small light intensity differences as evidenced by the nonlinear ON pathway as compared to the linear OFF pathway. Even if this nonlinear processing strategy leads to a loss of single-photon responses, this loss is almost fully compensated by the elimination of neural noise. This is evident from the observation that the nonlinear ON pathway and the

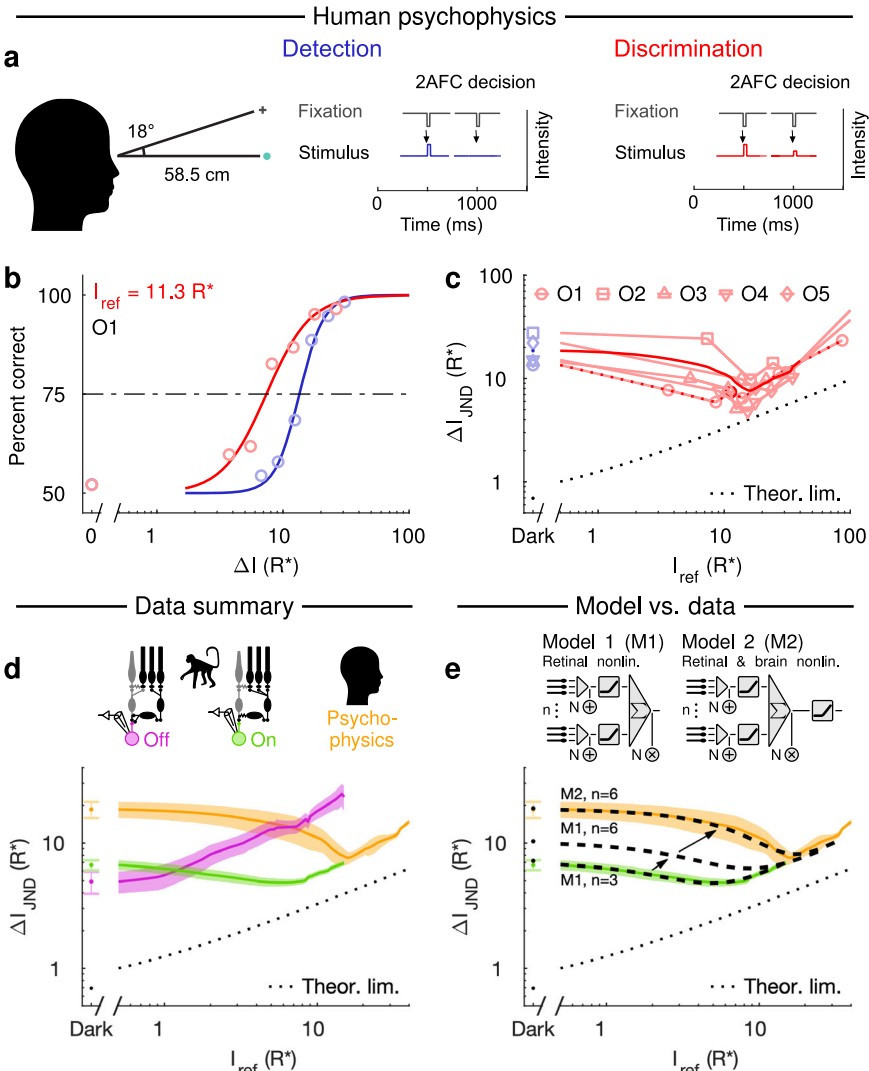

**Fig. 3 | The performance of human observers agrees with the new dim light detection model where performance is fundamentally limited by retinal thresholding nonlinearities along the ON pathway. a** The psychophysical experiment. In the detection task, the observer had to report which of the two intervals contained the flash. In the discrimination task, the observer had to report which of the two intervals contained the brighter flash. **b** Psychometric functions of a representative observer: the detection task (blue) and the discrimination task (red) for one reference flash intensity. **c** $\Delta I_{JND}$ (75% correct) as a function of the reference flash intensity (Dark for detection) for all observers. Thin lines correspond to individual observers, the dotted line denotes the example observer shown in (**b**), and the thick line denotes the population average. **d** Average $\Delta I_{JND}$ (mean ± SEM) as a function of the reference flash intensity for human observers (orange, $n = 5$ observers), OFF RGCs (magenta, $n = 11$ cells), and ON RGCs (green, $n = 54$ cells). **e** Model predictions (thick dashed lines) for ON RGCs (M1, number of subunits: $n = 3$, $\theta_1 = 2$; similar to the parameters used in Ala-Laurila and Rieke[10]), an ON RGC model with twice the number of subunits (M1, $n = 6$, $\theta_1 = 2$), and for a model with two thresholding nonlinearities (M2, $n = 6$, $\theta_1 = 2$, $\theta_2 = 2$): one in the retina and one downstream of the retina. Same average $\Delta I_{JND}$ (mean ± SEM) for a human observer (orange) and ON RGCs (green) as in (**d**). The dotted black line shows the theoretical performance limit set by quantum fluctuations in the number of stimulus photons. For the model robustness analysis and parametrization, see Supplementary Figs. 3 & 4. Source data are provided as a Source Data file.

linear OFF pathway perform almost equally well on a light detection task despite extra single-photon losses in the ON pathway (see also Ala-Laurila and Rieke[10]). Thus, visual perception of single photons has likely not been the central optimization goal during evolution. Instead, human vision approaches the theoretical limit of physics in discriminating dim flash intensities by utilizing a strategy where neural noise is eliminated at the expense of single-photon responses to optimize contrast coding in the retinal ON pathway.

Third, single-photon responses are separated from noise by thresholding nonlinearities in retinal circuits. The first synapse of the rod bipolar pathway thresholds rod signals to eliminate neural noise and many single-photon responses[28]. Since individual rods very rarely absorb overlapping photons for just-detectable inputs, this first nonlinearity functions as a linear loss mechanism for single-photon responses that is shared for ON and OFF pathways.

The last synapse along the ON pathway, on the other hand, requires a coincidence of two or more single-photon responses within a collection of several thousand rods[10], and this nonlinearity shapes retinal output signals. We add to this picture here by identifying an additional nonlinearity operating in a similar manner downstream of the retina. Such a post-retinal thresholding mechanism in the brain is similar to previous findings of nonlinear signal processing in thalamic neurons at higher light levels[29–31]. Together, these results show that a key neural strategy of high-fidelity coding of sparse signals is implemented by a combination of thresholding mechanisms operating at different levels of convergence of the neural circuits together with a large amount of pooling. It is likely that this general strategy and architecture are in use across other circuits and sensory modalities requiring sensitivity that approaches the limits of physics.

## Methods

### Data collection: retinal ganglion cells

Primate retinas were used as a proxy for human retinas since harvesting their dark-adapted ex vivo retinas at the uttermost sensitive conditions is more feasible than human retinas. It has also been recently shown that the spatiotemporal response properties of human and non-human primate ON and OFF parasol cells are very similar[32]. All recordings were from ON or OFF parasol retinal ganglion cells in the primate *Macaca nemestrina* (n = 11; 10 females and 1 male) and *Macaca fascicularis* (n = 7; 2 females and 5 males) retina (eccentricity > 30°) at the age of 4–21 years. Sex was considered in the study design to the extent that tissue was acquired from both female and male animals. Data were acquired with the Symphony data acquisitions system (https://symphony-das.github.io/) using a Multiclamp 700B amplifier and ITC-18 A/D board. The stimuli consisted of 10–20 ms flashes (5–12 different intensities and 20–140 repetitions per intensity) of a uniform circular light spot (diameter 560 µm), delivered from blue and/or green light-emitting diodes (LEDs; peak wavelength $\lambda_p$ at 460 nm and 510 nm). Visual stimuli were generated using the Stage software package (https://stage-vss.github.io/). The procedures have previously been reported[10]. All experiments were done in accordance with the guidelines for the care and use of animals at the University of Washington and all procedures were approved by the University of Washington Animal Care and Use Committee.

### Data collection: psychophysics

**Observers.** Five observers (three males (O1, O3 & O4), and two females (O2 & O5), aged 19–29 years) participated in the study. Sex was considered in the study design to the extent that both (self-reported) female and male participants were recruited. Observer O1 was one of the authors; the rest were naïve to the purposes of the study and received a small monetary compensation. All observers had normal uncorrected vision. This was validated by an ophthalmologist testing all observers for visual acuity and visual field sensitivity. All observers had normal monocular visual acuity (range 20/25–20/16) and no significant visual field defects within the eccentricity (<20°) were observed in a perimetry visual field test (Octopus 900, Haag-Streit Diagnostics, Switzerland). The study was conducted in accordance with the principles of the Declaration of Helsinki and the guidelines of the University of Helsinki ethical review board, which also approved the study. The participants signed a written informed consent.

**Stimuli.** The stimulus was a homogeneous circular light spot (diameter 1.17°≈315 µm on the retina; $\lambda_p$ = 500 nm) presented for 20 ms at 18° eccentricity in the lower visual field (superior retina). The fixation stimulus was a dim red ($\lambda_p$ = 680 nm) cross-hair with a diameter of 0.59° (157.5 µm) and a bar width of 0.098° (26.2 µm). The viewing distance was 58.5 cm, which is close to the expected dark focus distance for young adults[33].

**Apparatus.** The stimulus was produced by a combination of an LED (AND520HB, $\lambda_p$ = 466 nm), an interference filter (Edmund Optics, $\lambda_p$ = 500 nm, FWHM 10 nm), ND filters, a 0.8 mm aperture (to produce a more point-like source), a diffuser, and a circular aperture. The intensity of the LED was adjusted at the beginning of each experimental session with ND filters to produce a suitable stimulus range. During the experimental sessions, the light intensity and the flash duration were set using a National Instruments USB-6343 (National Instruments, Austin, TX, USA) DAQ and a custom LED controller. Nonlinearities in the stimulation system were corrected for by measuring the system's input-output function with a UDT Instruments S471 optometer and a UDT 268 R Photodiode (Gamma Scientific, San Diego, CA, USA), and by applying the inverse function to the intensity range.

The fixation stimulus was produced by a combination of an LED (AND180HRP), an interference filter ($\lambda_p$ = 680 nm, FWHM 10 nm), and a cross-hair-shaped aperture. Data acquisition and stimulus delivery were controlled using MATLAB (version: R2013a) and the Data Acquisition Toolbox.

**Procedure.** The threshold for discriminating between two weak flashes or a flash and darkness (detection) was measured with a 2AFC method of constant stimuli procedure. Each trial proceeded as follows (see Fig. 3a): The fixation stimulus blinked (i.e., turned off for 100 ms) four times, with a 500 ms period. A flash was presented at 18° eccentricity on the third and fourth fixation blinks, and the observer's task was to indicate during which blink the flash was stronger. For each run of trials, one of the flashes always had the same reference intensity ($I_{ref}$), whereas the other flashes had intensities within a predetermined range (7 steps), with the lowest intensity always being identical to the reference intensity. Consequently, when the reference intensity was 0, one of the flashes was always blank. The purpose of the four fixation blinks was to reduce temporal uncertainty.

One measurement session lasted approximately 1.5–2 h. In the beginning, the observer adapted to total darkness for 30 min. Then the observer performed about 500 trials at their own pace. The length of the session and the precise number of trials depended on the observer's pace and the level of subjectively perceived fatigue. Each data point in psychometric functions (such as shown in Fig. 3b) represents 144–228 trials for $I_{ref}$ = 0 and 74–140 trials for $I_{ref}$ > 0. The condition $I_{ref}$ = 0 was recorded twice for each observer, both at the beginning and at the end of the study, to reveal possible learning effects. The largest difference between the thresholds from the two measurements was 0.04 log units, which is negligible in comparison to the effect of reference intensity. The data from the two detection measurements has thus been pooled.

### Light intensity conversions

In order to compare stimulus intensities between RGC measurements and psychophysics, stimulus intensities were first converted into photoisomerizations per rod per second (R*/rod/s) and later to photoisomerizations per RGC (R*/RGC) or per retina (R*).

**Psychophysics.** For psychophysics data, the initial conversion to R*/rod/s was done as follows. Firstly, the stimulus power ($P_{stim}$), measured with a photodiode, was converted to a corneal photon flux density ($F_{cornea}$) as:

$$F_{cornea} = \frac{\lambda}{hc}\frac{P_{stim}}{A_{sensor}},\qquad(1)$$

where $\lambda$ is the wavelength of the stimulus light (500 nm), $h$ is Planck's constant, $c$ is the speed of light, and $A_{sensor}$ is the area of the photodiode (1 cm²). Secondly, the corneal photon flux density was converted to a retinal photon flux density ($F_{retina}$) by:

$$F_{retina} = \frac{F_{cornea}A_{pupil}}{A_{retina}}\tau_{media},\qquad(2)$$

where $A_{pupil}$ is the area of the pupil, $A_{retina}$ the area of the projected stimulus on the retina (0.077 mm²), and $\tau_{media}$ is the ocular media factor (about 45%[9]). $A_{pupil}$ was measured from video recordings of the dark-adapted observers (44.6, 56.4, 32.5, 53.2, and 49.5 mm² for O1, O2, O3, O4, and O5, respectively) while they fixated on the fixation stimulus in darkness (infrared illumination), whereas $A_{retina}$ was computed from the stimulus size in visual angles (diameter 1.17°) using the conversion factor from visual angle to retinal subtense at 18°

eccentricity (268 μm per °)[34]. Lastly, the photoisomerization rate per rod ($R_{human}$) for human observers was obtained as:

$$R_{human} = F_{retina} A_c^{human}, \qquad (3)$$

where $A_c$ is the collecting area of rods, obtained from:

$$A_c = \pi \left(\frac{d}{2}\right)^2 \left(1 - 10^{-\epsilon L}\right)\gamma, \qquad (4)$$

where $d$ is the diameter of the rod outer segments (2.27 μm at 18°)[35], $L$ is the length of the rod outer segment (42 μm)[36], $\epsilon$ is the specific absorbance (0.019 μm$^{-1}$)[37], and $\gamma$ is the quantum efficiency of rhodopsin (0.67)[38]. The values above resulted in a collecting area for human rods of 2.28 μm$^2$.

**Stimulus intensity in R\*.** Flash intensities ($I$) for the psychophysics data are given in R\* (per retina). These values were obtained by multiplying the photoisomerization rate per rod ($R_{human}$) by the stimulus duration (20 ms) and the total number of rods (10,325) beneath the area covered by the stimulus (rod density of 134,000 rods/mm$^2$ at 18°)[35].

**Ganglion cell recordings.** The intensities used in ganglion cell recordings were converted to photoisomerization rates per rod based on the spectral output of the LEDs, the spectral sensitivity of rods, and a collecting area ($A_c$) of 1.40 μm$^2$ ($d = 2$ μm and $L = 25$ μm)[38].

**Stimulus intensity in R\* per RGC.** Flash intensities ($I$) for RGCs are given in photoisomerizations per ganglion cell (R\*/RGC). These values were determined from the photoisomerization rates ($R_{RGC}$) as:

$$I = R_{RGC} N_{rods} t_{stim}, \qquad (5)$$

where $N_{rods}$ is the number of rods converging on the ganglion cell and $t_{stim}$ is the stimulus duration (20 ms). $N_{rods}$ was estimated from the size of the spatial receptive field (RF) by multiplying the effective area (integral over the areas of Gaussian weighted annuli up to the size of the stimulus) with the rod density. The size of the spatial RF, in turn, was quantified by presenting dim spots of various sizes for 250 ms and by fitting (least squares) a Gaussian-shaped RF to map the spot size to observed responses (i.e., assuming a Gaussian weighted linear spatial summation). The size of the spatial RF ($2\sigma$, where $\sigma$ = standard deviation of the Gaussian) was always found to extend beyond the imaged dendritic tree (see Supplementary Fig. 1). This observation is in line with the notion of a spatial spread of rod signals in the rod bipolar pathway as well as with previous literature showing ~2–3 times larger physiological RFs as compared to the morphological RFs at scotopic light levels[39]. We quantified the scaling factor between the size of the spatial RF and the dendritic tree by fitting an ellipse to the edge of the dendritic tree and by comparing the mean axis length of the ellipse to the size of the Gaussian-shaped RF. The scaling factor was not significantly different between ON and OFF parasols ($P = 0.36$, Welch's $t$-test; Cohen's $d = 0.61$), and we pooled all data to get a joint mean scaling factor of 2.1. The final estimate of $N_{rods}$ was thus obtained by combining the scaling factor, the size of the dendritic tree (diameter: 190 μm at 30°)[40], the rod density (110,000 rods/mm$^2$ at 30°)[41], and the stimuli size (560 μm). This resulted in a final estimate of 6880 rods per RGC. The given values at 30° are not representative for all recorded RGCs, but the estimate of $N_{rods}$ is: the convergence remains fixed as the rod density and dendritic tree size change with increasing eccentricity[42].

## Data analysis
We quantified discrimination performance in the retina using an ideal observer model that considered both the light response and the tonic firing rate when performing a 2AFC task[10,43]. The recorded spike response was condensed into a single number ($r$) for each epoch by computing the inner product of the binned (10 ms) spike counts with a discriminator over a 500 ms long interval. The discriminator was defined as the mean response difference to the reference intensity ($I_{ref}$) and all higher intensities. The ideal observer evaluated the discriminability as a function of flash intensity by comparing the response distribution (the $r$ values from all epochs) to a reference intensity ($I_{ref}$) with those obtained at higher intensities ($I_{ref} + \Delta I$). Performance on the detection task thus corresponded to the special case where $I_{ref}$ equaled zero, and for this special case, the responses were calculated from spontaneous activity (ON RGCs: $1.0 \pm 0.3$ Hz; OFF RGCs: $22.6 \pm 3.6$ Hz). The fraction of correct choices by the ideal observer was evaluated as:

$$P_{correct} = 0.5 P\left(r_{Iref} = r_{Iref + \Delta I}\right) + P\left(r_{Iref} < r_{Iref + \Delta I}\right) \qquad (6)$$

where $r$ denotes the response. The just noticeable difference was taken as the increase in flash intensity ($\Delta I$) that corresponded to $P_{correct} = 75\%$. This value was obtained by fitting (least squares) a modified sigmoidal Hill function to the data and by determining the intensity where the fit reached 75% (see Fig. 2c, d). The Hill function was defined as:

$$P_{correct}(I) = P_{min} + (P_{max} - P_{min})\frac{I^n}{I^n + K^n}, \qquad (7)$$

where $P_{min}$ is the chance level ($= \frac{1}{2}$), $P_{max}$ is the best possible performance ($\leq 1.0$), $I$ is the stimulus intensity, $K$ is the intensity at $P_{correct} = \frac{P_{min} + P_{max}}{2}$, and $n$ is the slope. For the ganglion cell data, the number of available data points decreased for each consecutively higher reference intensity, as responses were only available from a fixed number of flash intensities. We solved this by modeling additional response distributions for intermediate flash intensities using normal distributions with the mean and variance fixed to interpolated mean and variance values from the real data (see Supplementary Fig. 2). The modeled response distributions were always interpolated between the reference intensity and the subsequent intensity, and the number of modeled distributions was determined so that the Hill functions could be fit to at least five data points. The psychophysics data was analyzed in an equivalent manner, with the Hill functions fit to the response probabilities directly (see Fig. 3b). Ganglion cells were included in the main analyses if they fulfilled similar sensitivity criteria as established earlier[10]: RGCs had to generate an average spike difference of four to five spikes in response to a brief flash producing 0.001–0.002 R\*/rod as compared to the baseline firing rate.

## Modeling
RGC and psychophysics dippers (Fig. 2f and Fig. 3c) were modeled (Fig. 3e) by mapping discrete input events (isomerizations) to discrete outputs (responses) using a subunit model. Thus, this model is not meant to capture the time course of responses but instead focuses on the impact of thresholding nonlinearities. The model considers a single set of excitatory subunits, as we have shown previously that the excitatory synaptic inputs and the spike outputs of ON parasol RGCs share similar sensitivity and thresholding nonlinearities[10]. The stimulus (signal) was distributed uniformly over all subunits and the responses were analyzed in the same way as the recorded data, the only difference being that the $P_{correct}$ values could be calculated from the modeled output distributions directly. These output distributions were

obtained by (1) pooling (adding) Poisson distributed signals and noise within each subunit, (2) passing the result through a rectifying non-linearity, (3) summing up the contribution from each subunit, (4) including multiplicative noise, and (5) by thresholding the sum. Summation of the outputs from each subunit was implemented via convolutions, and the multiplicative noise was included by letting the summed subunit output set the mean response for a sequential Poisson process via a gain term (see Field et al.[7]). In total, the model thus had five parameters: the magnitude of the additive noise, the subunit nonlinearity's threshold, the number of subunits, the gain parameter for the multiplicative noise, and finally, the threshold for the post-retinal nonlinearity. In all cases, the optimal values for free parameters were found by minimizing the mean squared error between the model's dipper function and the target dipper function (from RGC or psychophysics data). The effects of varying each parameter separately are shown in Fig. 3e (number of subunits) and in Supplementary Fig. 3 (remaining parameters). Supplementary Fig. 4 shows the model robustness when changing each key model parameter either by doubling or halving their values. The main conclusions of the paper – namely that the RGC dataset and psychophysical datasets can be bridged within the "new model" (M2) and not within the "classical" model (linear retina model) – are robust across any reasonable parameter space.

### Theoretical limit

The theoretical limit (dotted line in figures) corresponds to the modeled output when no noise or thresholds are included. That is, it corresponds to linear processing in the absence of noise and thus reflects $\Delta I_{JND}$ for detection and discrimination due to inherent Poisson fluctuations in the stimulus.

### Statistical analysis

Reported values are mean ± SEM. MATLAB (version: R2017b) was used for all statistical analyses, and all reported tests were two-tailed. The statistical significance and effect size of the ON-OFF detection and discrimination threshold difference were determined by conducting a Welch's $t$-test (and Cohen's $d$) on the cells' average detection and discrimination thresholds (averaged over all $I_{ref}$ values). The statistical significance and effect size of the psychophysics dips were determined separately for each observer by conducting a Student's $t$-test (with Bonferroni correction) and Cohen's $d$ on the $K$ parameters (75% correct) of the Hill functions fitted to the detection and discrimination data (solid lines in Fig. 3b). For observers O1, O2, O3, O4, and O5, respectively, the most significant $t$-score was 13.72, 17.98, 4.21, 12.12, 11.19; $P$-values <0.001, <0.001, 0.007, <0.001, <0.001; Cohen's $d$-values 7.33, 9.61, 2.25, 6.48, 5.98. Pooled SD was used in calculating Cohen's $d$.

### Reporting summary

Further information on research design is available in the Nature Portfolio Reporting Summary linked to this article.

## Data availability

All data generated and analyzed in this study are available in the Figshare database: https://doi.org/10.6084/m9.figshare.25737099. Source data are provided with this paper.

## Code availability

For the code, see the Zenodo repository: https://doi.org/10.5281/zenodo.10459936.

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

## Acknowledgements
We thank Drs. Kristian Donner, Greg Field, Kai Kaila, Gabriel Peinado Allina, Greg Schwartz, and Mark Georgeson for excellent comments on the manuscript; Matthew Dunkerley, Sathish Narayanan, and Mark Cafaro for the design of the data acquisition software. Support was provided by the Academy of Finland (296269 to P.A.-L.); the Aalto Brain Centre (J.W.); Svenska kulturfonden (J.W.); The Finnish Society of Sciences and Letters (J.W.); The Ella and Georg Ehrnrooth Foundation (M.K.); Aalto Centre for Quantum Engineering (CQE) grant (J.T.); University of Helsinki Brain & Mind Doctoral Programme funded position (J.T.) and the NIH (EY 028111 to F.R.). Human silhouette in Fig. 3 and Supplementary Fig. 2 retrieved from Vecteezy.com. We thank Juha Haapala for preparing the monkey silhouette used in Figs. 2, and 3.

## Author contributions
F.R., J.W., M.K., P.A.-L. designed experiments, F.R. and P.A.-L. collected RGC data, M.K., A.L., J.T. collected psychophysics data, F.R., P.A.-L., J.W., M.K., D.T., J.T. analyzed data, F.R., J.W., M.K. & P.A.-L. wrote the paper.

## Competing interests
M.K., J.W., J.T., A.L., D.T., F.R. declare no competing interests. P.A.-L. is a founder and shareholder of Quantal Vision Technologies.
