## [Peer Review File · Nature Communications]

Primate retina trades single-photon detection for high-fidelity contrast encodingREVIEWER COMMENTS

Reviewer #1 (Remarks to the Author):

Kilpeläinen and colleagues here progress a line of research on visual processing at low light levels, building on a key finding from 2014 (Ala-Laurila and Rieke, *Curr Biol*, cited in the present study), that there is an absolute sensitivity difference between ON- and OFF-parasol retinal ganglion cells, with OFF cells showing higher absolute sensitivity and greater noise levels than ON cells do. (Parasol cells are the beginning of a visual pathway serving motion perception and form vision at low image contrasts and low intensities). Here the authors use signal processing theory to show that although the absolute sensitivity of ON cells is lower than that of OFF cells, the ON cells are better at discriminating intensity levels of flashes presented on scotopic backgrounds. This physiological result will be of broad interest to sensory neuroscience, for its broad relevance sensory processing as well as specific relevance to understanding how we can get around at very low light levels. The authors use a similar set of stimuli to assess detection and discrimination performance of five human subjects and show results consistent with presence of thresholding nonlinearities in human intensity discriminations. This second result is tied to the performance of ON parasol ganglion cells by virtue of common "dipper" discrimination performance shifts close to threshold. The psychophysical result should also be of interest to the (admittedly rather small) scotopic vision specialists and the attempt to link physiology with psychophysics is interesting and compelling with some minor concerns given below.

The work appears of excellent scientific quality as expected from Ala-Laurila and Rieke groups, is clearly and cogently argued, and the claims are well supported by the data. The psychophysical methods are presented in enough detail to enable the work to be reproduced, and the physiological methods have already been described in detail. It could be argued that important limitations to the "Classical" linear model shown in Fig. 1a are already well established (by the authors among others, as explained on page 2) but this concern is more than offset by the strong parallels between single cell physiology and human performance given in this fine study.

Minor

P1 line 20 "Surprisingly ..." [here and elsewhere one finds advertorial adverbs sprinkled without succinct explanation. Why should the reader be surprised? This sentence, and the manuscript, would be better off without them in this reviewer's opinion.]

P2 line 2 "Strikingly ..." [ditto]

P4 line 25. "We tested ... excellent fit ... " [manipulating values of a four-parameter model has limited explanatory or predictive power, and alignment of curves does not logically imply causation. Some words of caution would be welcome here]

P5 line 22 "It remains to be seen ... " it would be churlish to insist on such experiments (one presumes, but is not told, that the physiological data set did not include such measurements), but are some pilot observations not possible on human subjects? This could yield interesting results and improve the general interest of the study. The suggestion is for the authors' consideration.

P5 line 27 "responses to individual photons interact " [? this is a cryptic statement, please explain]

P6 line 18 [publications from Barry Lee's laboratory in last millennium first implied presence of post-retinal filters in detection performance. For the authors' consideration]

P10 line 21 Why do the authors pick $2[\sigma]$ as a definition? This result shows that the definition does not match the data. Other work likewise shows closer match of d_f diameter to r_f center diameter (albeit under photopic conditions) -- see e.g. Fig 19, *Physiol. Rev.* 1991, Wassle and Boycott.

P11 line 15 What were the spontaneous levels, please?

Fig 1. The schematic is largely clear, but presence of question marks is unexplained and unenlightening.

Fig. 3 A more gender-neutral human silhouette might be preferred.

-- Brief answers to review form questions below.

- What are the noteworthy results?

Demonstration that ON-pathways serve intensity discrimination performance at low light levels.

- Will the work be of significance to the field and related fields?

Yes

- Does the work support the conclusions and claims, or is additional evidence needed?

Conclusions and claims are supported.

- Are there any flaws in the data analysis, interpretation and conclusions? - Do these prohibit publication or require revision?

No obvious flaws are evident

- Is the methodology sound? Does the work meet the expected standards in your field?

Yes

- Is there enough detail provided in the methods for the work to be reproduced?

Yes

Reviewer #2 (Remarks to the Author):

The study by Markku Kilpelainen and colleagues explores the mechanisms underlying detection and discrimination of dim lights near the threshold of human vision. The authors set up the problem as a conflict between past psychophysical studies and recent findings (including studies by the last two authors), where the retina's ability to detect single photons and its consequences for perceptual tasks under very dim light conditions remain ambiguous. Using a 2AFC task, the authors show that ON parasol RGC responses are more sensitive to discrimination between dim flashes than those of OFF parasol RGCs, which arises due to nonlinear processing of signals associated with single photon absorptions by ON parasol RGCs. The authors further show using a subunit model of the retina, that difference in discrimination performance between human observers and those estimated from responses of ON parasol RGCs, can be explained by an additional nonlinearity located downstream of the retina.

This is a well-designed study with a clear rationale. The methods are sound and analysis of data is thorough and detailed. While the asymmetry between ON and OFF parasols' processing of scotopic stimuli has been established, linking this asymmetry to behavioral detection limits reached by humans, and providing clues for the signaling mechanisms that allow discrimination near absolute threshold, are important findings. However, the results need to be better contextualized in the light of past studies and limitations need to be addressed, for broader adoption and appreciation of the study.

Major comments:

1. While I believe that these findings would be valuable to the community, I have some concerns regarding the novelty and interpretation of the results. First, the last authors themselves showed in an earlier study using a 2AFC task, that ON parasols strongly rectify signals unlike OFF parasols (Ala-Laurila, 2014). In the same study, the authors showed and that OFF parasols are more sensitive to dim flashes and flash discrimination is asymmetric between the two pathways. The novelty of the current study is that it provides a contextual link between these response properties and human perception near visual threshold. However, the discussion doesn't sufficiently describe these past findings upon which the current work is based. Second, the subunit model described here doesn't take into consideration the effects of feedback inhibition (Grimes, 2015) or crossover inhibition (Cafaro, 2013) by amacrine cells other than AII, which can regulate gain of signal transmission at RBC synapse. Coupling between rods and cones at dim lights can also provide an alternative high gain pathway for rod signals initiated by single or multi-photon absorption events. What are the relative contributions of these factors to spike generation at these light conditions? Third, the perceptual task presumably depends on retinal output from not just ON parasol pathway, which might be the dominant pathway, but on all retinal pathways that carry information about the stimulus (dim flashes). Without knowing how these other pathways contribute and how the signals from those pathways are processed in downstream circuits in the LGN and V1, it seems premature to conclude the current framework fully describes the mechanisms governing human perception at threshold.

2. A key findings of the study is that adding a nonlinearity downstream of the retina allows recapitulation of the performance curve (aka, the location and amount of dip) obtained from psychophysical experiments. While exploring this nonlinearity may not be achievable in the current setup, the importance of this nonlinearity cannot be overstated. For example, when rods die in inherited retinal diseases, RGCs can be targeted for stimulation using optogenetics or prosthetics. In the absence of photoreceptor inputs, the ability to detect and discriminate dim lights would inevitably depend on downstream circuits and their nonlinearities. Studied such as Kremkow et al. 2014, suggest that such nonlinearities may be engaged in ON thalamic pathways but not in OFF thalamic pathways. It would be nice if the authors could discuss these in the context of the present study.

3. From Figure 2 e, f, it appears that OFF parasols, on average, performed slightly better at detection than ON parasols. The threshold intensity for detection is $<10 R^*/RGC$ for all OFF parasols, while $>10 R^*/RGC$ for several ON parasols. This is also observed in Supp Fig 2, by the leftward shift of the 2AFC detection curves for the OFF parasol RGC. These results run contrary to the authors' claim that both these cell types have similar detection performances. I wonder if the population variance can explain the discrepancy. Is the small dark blue circle the population mean? If it is not, then the authors should include the population mean the SEM. If it is, then the authors should increase the size of the maker and include the SEM for the population. These values should also be included in the figure caption.

Minor comments:

1. Study participants are denoted by S1, ..., S5 (pg 9, line 19), flash strengths are denoted by S_1, S_2,... (pg 16, line 11), and supplementary figures are enumerated as S1-S3 (pg 6, line 26). This can be confusing for the readers. I suggest using different letters to denote flash strength and observer identity.

2. On page 5, the authors test the extent to which difference between psychophysical responses and ON parasol responses depend on number of activated RGCs. The authors do this by increasing the number of subunits from 3 to 6. What is the reason for doubling the number of subunits? Does the size difference between the stimulus spot projected at 30deg eccentricity and the receptive field correspond to 3 additional subunits? This needs explanation.

4. Nonlinear processing of visual signals in contrast discrimination tasks has been shown to

produce dipper-like curves (Legge & Foley, 1980; Nachmias & Sansbury, 1974). The authors should consider citing these studies. The authors should also cite references for the subunit model (Hichstein, 1976; Shapley, 1979; Crook, 2008).

5. The authors should cite the study by Soto et al., 2020, which shows evidence for conserved midget and parasol pathways in human retina.

6. What are the parameters for high vs low noise and high vs low threshold in Supplementary Fig. 3? Is the noise regime set by the quantum efficiency of photon absorption?

7. The authors state that "All observers had normal uncorrected vision" (pg 7, line 24). Were Visual Field Test or Perimetry Test done to confirm that the subjects had normal vision?

8. The statement "our results indicate that response to individual photons interact, providing evidence for the perceptual relevance of detection of coincidence photons" is confusing because it does not answer the question whether humans can perceive single photons. In fact this paragraph reads that single photons are absorbed by rods but downstream processing makes only (or largely) information about coincident photons available to the brain for perception. Also, "humans can see a photon" phrase can be misconstrued. Seeing is an experience that is shaped by factors unrelated to the visual task. I think this paragraph should be revised to make the interpretation clear.

Reviewer #3 (Remarks to the Author):

The authors aim to answer the question of whether human visual processing at extremely low light levels is linear, or whether there is a nonlinear threshold in the retina before visual signals reach the brain. This manuscript presents new evidence that may distinguish between these two alternatives. The primary approach is to use measured performance in detection vs. discrimination tasks to distinguish between the models, which make different predictions for performance in these two types of tasks. In particular, the authors report two experiments: an in vitro study using output signals of the primate retina as a proxy for human retinal outputs, and a two-alternative forced-choice (2AFC) behavioral study with human subjects. The authors conclude that both experiments support their hypothesis that there is a nonlinear threshold in the human retina, and that there is additional thresholding in the brain.

This is an interesting, timely, and well-conceived study. The question of whether there is a nonlinear threshold in the retina at extremely low light levels is important. In particular, it is relevant to the question of whether humans can detect single photons, which has been a subject of debate essentially since photons were discovered, and has implications for possible future quantum science experiments. The hypothesis that there may be a photon coincidence requirement (nonlinear threshold) somewhere in retinal processing, and that single-photon detection is therefore not possible, has been an important aspect of this debate. Evidence in support of this hypothesis therefore has a major impact on the study of human vision at the single-photon level. The methods used in this article are appropriate and provide new and useful experimental data for answering the main question. In most cases the data support the conclusions, although some conclusions may be overstated (see detailed notes).

There are two main issues that should be addressed. First, on page 2, the authors mention that recent results from primate retinas and mice have suggested that the linear model of visual processing may be incorrect. It seems that the primary goal of the current work is to find out whether these results also apply to humans—an important question! The manuscript should include a statement along the lines of "A study that combined retinal signal measurements with behavioral studies in mice found X, and measurements with primate retinas suggested that something similar is also happening in primates. The mouse experiment can't be replicated with human subjects, so in order to know whether this same effect happens with humans, we need to know Y and Z. Our measurements with primate retinas provide Y and our human behavioral

experiments provide Z, therefore we can conclude that what was observed in mice is also likely happening in humans.” Without this explicit context, I believe it will be difficult for a typical reader to understand whether these results really prove the authors’ broader hypothesis.

Second, the authors must directly address how their results are or are not consistent with behavioral evidence for single-photon detection by humans, in particular Tinsley et al. Tinsley et al claimed to have evidence for single-photon vision in a 2AFC task, so this manuscript’s claim that single-photon vision is impossible is in direct conflict with that result. Observing an effect tends to be more convincing than arguing why it is impossible. It is not necessarily these authors’ responsibility to explain why the Tinsley study found different results—and there are many possibilities; for example, it had low statistical power and didn’t have a control group—although if they have insight, it would certainly add to the scientific discourse on the subject. But if they are going to make the case that the “decades-old question about whether humans can see a single photon” has now been answered, it requires more discussion to put that conclusion in context, especially in light of contradictory evidence. The disagreement with other recent studies, e.g. Tinsley et al, is one reason why this research is significant.

Finally, the statistical analysis is missing a discussion of uncertainty in the measured values, particularly the threshold θ , including sources of uncertainty from the light intensity conversions. A best fit value of θ was determined by fitting a model, but would other values of θ also be consistent with the experimental data within uncertainty? Some plots showing experimental data also lack error bars.

Please see detailed comments below.

Page 2, Line 8: I disagree somewhat with the characterization of the two models as “classical” and “new” – I think neither model was previously totally accepted or supported by evidence, which is in fact a justification of the importance of the current study. However, I may be wrong. My suggestion would be to call them “linear” and “nonlinear” models.

Page 2, Line 13: Suggest briefly introducing ON and OFF pathways for the reader – this is not a specialized publication in psychophysics/vision science.

Page 2, Line 20: “in which perception [near threshold]” relies on retinal outputs ...” Or is it really perception at all light levels?

Figure 1

- a-b: Why is there an arrow directly from spikes to behavioral output in 1b? I would think all behavioral output needs to involve the brain?
- Caption Line 4 – Typo “reference”
- c-d: It would be helpful to give the values of N, S1, S2, and S3 in this figure (it’s possible to tell from the plots, but the values would still be nice). Also, if the conclusion is that the threshold is 2 photons, why does this figure show an example where the threshold appears to be 6?
- The combination of linear and nonlinear scales makes it hard to tell where the “dip” occurs and how it relates to the threshold.

Page 5, Line 3 – Additive and multiplicative noise levels were fit as model parameters, but I can’t find the actual noise values in the manuscript. Were reasonable noise levels required to achieve the best agreement with data?

Page 5, Line 18 – I believe the language here is somewhat more conclusive than is justified by the evidence presented. This work shows that a plausible model is in good agreement with experimental data, which is a very reasonable way to approach the problem, but is not quite direct evidence comparable to the mouse study in Reference 10. Instead of “human behavior at absolute threshold relies on retinal output signals provided by ON RGCs”, I would suggest “our results support the hypothesis that human behavior at absolute threshold relies on retinal output signals provided by ON RGCs”. Similarly, say “This result shows that retinal ON and OFF pathways likely carry out distinct functional roles ...”

As discussed in the introductory comments to this referee report, a clearer explanation of how this work fills in the gaps of previous primate and mouse studies would help the reader judge the strength of the conclusions. The authors should also discuss alternative explanations of their results or other models that could explain the data, if there are any.

Page 5, Line 26: "Our results answer the decades-old question about whether humans can see a single photon" – with a setup like that, the reader is expecting a clear "yes" or "no" in the rest of the sentence. Also, as discussed in introductory comments to this referee report, there must be a direct discussion of how these results are/are not in conflict with Tinsley et al.

Page 6, Line 3: Suggest changing to something along the lines of "Thus, seeing single photons has likely not been the central optimization goal during evolution." I don't think there is any direct evidence for "evolutionary optimization goals" in this study.

Page 9, line 17 – If the value of the ocular media factor is important to the conclusions drawn (for example, the value of the threshold) the authors should note that it is very approximate and known to vary between individuals. (Reference 8 states: "All of these factors vary considerably between individuals, so any calculations we make with them will be approximate.") This uncertainty has been a major limitation of studies of single-photon vision using classical light sources. If the exact value of the ocular media factor is not important to the conclusions, please say so. If it is, the authors should estimate how it contributes uncertainty to their estimate of the threshold.

Page 13, Statistical Analysis – Include discussion of uncertainty in the measured values, particularly the threshold θ .

Reviewer comments:

Reviewer #1: Overview

Kilpeläinen and colleagues here progress a line of research on visual processing at low light levels, building on a key finding from 2014 (Ala-Laurila and Rieke, *Curr Biol*, cited in the present study), that there is an absolute sensitivity difference between ON- and OFF-parasol retinal ganglion cells, with OFF cells showing higher absolute sensitivity and greater noise levels than ON cells do. (Parasol cells are the beginning of a visual pathway serving motion perception and form vision at low image contrasts and low intensities). Here the authors use signal processing theory to show that although the absolute sensitivity of ON cells is lower than that of OFF cells, the ON cells are better at discriminating intensity levels of flashes presented on scotopic backgrounds. This physiological result will be of broad interest to sensory neuroscience, for its broad relevance sensory processing as well as specific relevance to understanding how we can get around at very low light levels. The authors use a similar set of stimuli to assess detection and discrimination performance of five human subjects and show results consistent with presence of thresholding nonlinearities in human intensity discriminations. This second result is tied to the performance of ON parasol ganglion cells by virtue of common "dipper" discrimination performance shifts close to threshold. The psychophysical result should also be of interest to the (admittedly rather small) scotopic vision specialists and the attempt to link physiology with psychophysics is interesting and compelling with some minor concerns given below.

The work appears of excellent scientific quality as expected from Ala-Laurila and Rieke groups, is clearly and cogently argued, and the claims are well supported by the data. The psychophysical methods are presented in enough detail to enable the work to be reproduced, and the physiological methods have already been described in detail. It could be argued that important limitations to the "Classical" linear model shown in Fig. 1a are already well established (by the authors among others, as explained on page 2) but this concern is more than offset by the strong parallels between single cell physiology and human performance given in this fine study.

We thank the reviewer for this excellent summary of our manuscript. We are pleased to see that the reviewer points out that our results will be of broad interest in the field of sensory neuroscience and highlights the high quality of the work.

Minor Comments:

1. P1 line 20 "Surprisingly ..." [here and elsewhere one finds advertorial adverbs sprinkled without succinct explanation. Why should the reader be surprised? This sentence, and the manuscript, would be better off without them in this reviewer's opinion.]

Action points: We have now eliminated "surprisingly" on P1 line 22 in the revised manuscript and edited the presentation elsewhere, as well, such that advertorial adverbs have been eliminated and/or explained more carefully.

2. P2 line 2 “Strikingly ”...” [ditto]

Action points: We have now replaced “strikingly” in the sentence mentioned by the reviewer (currently P4, line 32) and in P5, line 20.

3. P4 line 25. "We tested ... excellent fit ... " [manipulating values of a four-parameter model has limited explanatory or predictive power, and alignment of curves does not logically imply causation. Some words of caution would be welcome here]

Arguments: We agree that we need to use more careful statements about the modeling results and justify the model’s robustness more clearly. The parameters allowing the model to fit the data within a physiologically constrained parameter space require the addition of a second nonlinearity. This is not evident in the original presentation since we did not have a demonstration of model robustness.

Action points: We toned down the statement “excellent fit” (P6, line 16). More importantly, we have now added a new supplementary figure (Supplementary Fig. 4) to demonstrate the robustness of the model against reasonable parameter variations and to provide a better perspective of the goodness of the model fit.

4. P5 line 22 "It remains to be seen ... " it would be churlish to insist on such experiments (one presumes, but is not told, that the physiological data set did not include such measurements), but are some pilot observations not possible on human subjects? This could yield interesting results and improve the general interest of the study. The suggestion is for the authors' consideration.

Arguments: This is a timely point. Our physiological datasets do not include decrement stimuli (neither in human psychophysics, nor on primate RGCs). We did not include such datasets in this paper because the classical question on the absolute sensitivity limit of vision is fully based on using light increments shown at the visual threshold as a stimulus. Thus, we fully agree with the reviewer that insisting on light decrement stimuli on RGCs and human psychophysics in this paper would not be reasonable. It would require more than doubling the experimental data sets. More importantly, we feel that including light decrement datasets in this paper would shift the discussion away from the original focus, where we link our findings to the decades-old question of what is the right conceptual framework to understand the neural mechanisms and retinal pathways that underlie perception of the dimmest light increments in darkness.

We are currently working towards the new direction proposed by the reviewer and report below some pilot observations following the reviewer’s suggestion (see Fig. 1). The experimental work requires a combination of human psychophysics and primate RGC recordings in matching conditions using light decrement stimuli, similarly, as the current study relying on light increments. A particularly important realization is that the dimmest light decrements (“*quantal shadows*”, see Westö et al., 2022), in contrast to light increments, have a stimulus-generation-related limit set by the intensity of the background light. One can always make a stimulus brighter but never darker than darkness (see Westö et al., 2022). One would, therefore, need to increase the stimulus duration, as compared to short flashes, to be able to

test whether human quantal shadow detection depends upon the retinal OFF pathway. Such stimuli have never been reported on primate RGCs to our knowledge. We have an ongoing project towards these goals, and we will publish the results in a full-scale study, as the data is collected. This includes both human psychophysics and primate RGC data as a model of human retinal responses close to the visual threshold. Fig. 1 below shows pilot data on a single dark-adapted human subject: It shows a comparison of light increment and decrement detection thresholds using both global homogenous light fields (large symbols) as well as local spots (spot size = 315 μm , 18° eccentricity). The red lines (dotted and dash-dotted) show the maximal light decrement stimulus as a function of the background light. Psychophysics measurements allow threshold measurements only when the detection threshold is lower than the maximal decrement stimulus strength (below the red line). Our data using such small spot stimuli show very similar results to a previous study (Short, 1966). In order to compare light increment and decrement detection in the light regime, where rod bipolar pathway mediates signaling in the retina ($< \sim 3 \text{ R}^*/\text{rod/s}$, Grimes et al. 2018), one needs to use large spatial and/or temporal domains such that the detection threshold is below the maximal light decrement. This is crucial for making comparisons between human psychophysics and retinal measurements relying on the most sensitive ON and OFF RGC types as proxies of the rod bipolar pathway readout. We show in Fig. 1 using global light fields that light increments and decrements have very similar detection thresholds even at background lights that are in the range of thermal isomerization rate of rhodopsin ($\sim 0.004 \text{ R}^*/\text{rod/s}$, Field et al. 2019). These pilot recordings with the data from Short (1966) test human vision on light decrement detection in the light domain, where retinal signaling is driven by the rod bipolar pathway. To test whether behavioral detection of the weakest light decrements relies on the retinal OFF pathway (i.e. to address the question we pose in the original manuscript), we need to record the response properties of the most sensitive ON and OFF RGCs on primate retina in matching conditions with human psychophysics. We need to use long enough stimulus durations ($> 100 \text{ ms}$) to be able to produce strong enough decrements to test the absolute sensitivity limit of increment and decrement detection in matching conditions.

Fig. 1. Human sensitivity thresholds for local (small markers) and global (large markers = full-

field stimulation in a full-field Ganzfeld) increment (blue) and decrement (red) stimuli. The black lines are fitted generalized Weber functions highlighting how the thresholds change as a function of the background intensity. The dotted red line denotes the maximal decrement possible for a 20-ms decrement stimulus (i.e., the stimulus magnitude obtained when using a perfectly dark stimulus). The dash-dotted red line denotes the maximal decrement possible for a 100-ms decrement stimulus. Decrement thresholds can thus only be measured psychophysically in backgrounds where the black line is below the dotted red line. At dimmer backgrounds, it is simply not possible to create a strong enough decrement stimulus. Comparisons of increment/decrement detection thresholds at the lowest light levels for local stimuli will require stimulus durations that are longer than 100 ms.

Action points: We have now edited the text in the paragraph (P7, starting on line 3) referring to these human experiments. Particularly, we describe that the experiments testing the detection limit of the dimmest light decrements (“*quantal shadows*”) requires a particularly careful stimulus design taking into account the fundamental limits set by the intensity of the background light. As described above, such a study requiring carefully matched experiments on monkey RGCs and human psychophysics, will deserve its own paper.

5. P5 line 27 "responses to individual photons interact " [? this is a cryptic statement, please explain]

Action points: We have now clarified this statement (P7, line 20): “*Our results here indicate that responses to individual photons in the retina sum nonlinearly, providing evidence for the perceptual relevance of detecting coincident photons*”³.

6. P6 line 18 [publications from Barry Lee's laboratory in last millennium first implied presence of post-retinal filters in detection performance. For the authors' consideration]

Arguments: Thanks for pointing out Barry Lee’s papers.

Action points: We have now added a reference to the relevant paper. (P8, line 13).

7. P10 line 21 Why do the authors pick 2σ as a definition? This result shows that the definition does not match the data. Other work likewise shows closer match of df diameter to rf center diameter (albeit under photopic conditions) -- see e.g. Fig 19, *Physiol. Rev.* 1991, Wassle and Boycott.

Arguments: We quantified the physiological receptive fields of RGCs by using Gaussian fits, where σ = standard deviation of the Gaussian. The measurements clearly show, that the RGCs integrate stimulation from an area wider than the dendritic field (see Supplementary Fig 1). As long as the width of the Gaussian is defined, we do not see any major difference in whether the RF sizes are defined based on one or two-sigma definitions. Those definitions unambiguously still define the same RF Gaussian function fit to represent the physiological RF. In other words, the function defining the RF remains the same, and 2σ is a pretty widely used measure for the width of the RFs. The match between physiological and anatomical RFs depends greatly on stimulus conditions. At photopic light levels, physiological and anatomical RFs are in better

agreement with each other than at scotopic light levels. This is due to a larger spatial spread of retinal signals at low light levels. For example, the paper that the reviewer points out (Wassle and Boycott, 1991) has been carried out in photopic light levels ($\sim 100\,000\text{ R}^*/\text{rod/s}$). At these light levels, cone bipolar cells mediate signals directly to ganglion cells. In dim lights, physiological RGC RFs are $\sim 2\text{--}3$ times larger than morphological RFs. Indeed, we also observe a nearly 2-fold increase in the physiological RFs of ON and OFF parasols as we change from photopic to low scotopic light levels. This is likely due to the spatial spread of signals as these traverse the retina via the rod bipolar pathway as compared to direct cone pathways at higher light levels.

Action points: We have now added the following sentences (P12, lines 10–15): “*The size of the spatial RF (2σ , where σ = standard deviation of the Gaussian) was always found to extend beyond the imaged dendritic tree (see Supplementary Fig. 1). This observation is in line with the notion of a spatial spread of rod signals in the rod bipolar pathway as well as with previous literature showing $\sim 2\text{--}3$ times larger physiological RFs as compared to the morphological RFs at scotopic light levels*”³⁹

8. P11 line 15 What were the spontaneous levels, please?

Action points: We have now added the spontaneous rates (P13, lines 5–8): “*Performance on the detection task thus corresponded to the special case where I_{ref} equaled zero, and for this special case, the responses were calculated from spontaneous activity (ON RGCs: $1.0 \pm 0.3\text{ Hz}$; OFF RGCs: $22.6 \pm 3.6\text{ Hz}$).*”

9. Fig 1. The schematic is largely clear, but presence of question marks is unexplained and unenlightening.

Arguments: We agree that the legend of our schematic needs a better explanation of the brain compartment of the models.

Action points: We have now edited the caption of Figure 1 and explained the meaning of the three question marks in panels **a** and **b** of the schematic (see P20–P21, Fig. 1 legend).

10. Fig. 3 A more gender-neutral human silhouette might be preferred.

Action points: We now use a gender-neutral human silhouette in Fig. 3.

Reviewer #2: Overview

The study by Markku Kilpelainen and colleagues explores the mechanisms underlying detection and discrimination of dim lights near the threshold of human vision. The authors set up the problem as a conflict between past psychophysical studies and recent findings (including studies by the last two authors), where the retina’s ability to detect single photons and its consequences for perceptual tasks under very dim light conditions remain ambiguous. Using a 2AFC task, the authors show that ON parasol RGC responses are more sensitive to discrimination between dim flashes than those of OFF parasol RGCs, which arises due to

nonlinear processing of signals associated with single photon absorptions by ON parasol RGCs. The authors further show using a subunit model of the retina, that difference in discrimination performance between human observers and those estimated from responses of ON parasol RGCs, can be explained by an additional nonlinearity located downstream of the retina.

This is a well-designed study with a clear rationale. The methods are sound and analysis of data is thorough and detailed. While the asymmetry between ON and OFF parasols' processing of scotopic stimuli has been established, linking this asymmetry to behavioral detection limits reached by humans, and providing clues for the signaling mechanisms that allow discrimination near absolute threshold, are important findings. However, the results need to be better contextualized in the light of past studies and limitations need to be addressed, for broader adoption and appreciation of the study.

We thank the reviewer for this thoughtful summary of our work as well as for complimenting the rationale and the design of our study. We appreciate the reviewer's statement highlighting the importance of our findings linking the asymmetry in ON and OFF parasol coding to human psychophysics at the sensitivity limit of vision and providing evidence of the neural mechanisms underlying discrimination performance near the absolute threshold. In the revised manuscript, we now take into account the reviewer's critical suggestions related to the clarity of the presentation in terms of contextualizing the findings in the light of past studies and their limitations. We believe that these revisions outlined below will help the broader audience to better assess the importance of this study. We have addressed this request particularly by revising the Introduction section of the study as well as the Discussion section. We also revised various parts of the Results section related to this comment, as outlined below.

Major comments:

1. While I believe that these findings would be valuable to the community, I have some concerns regarding the novelty and interpretation of the results. First, the last authors themselves showed in an earlier study using a 2AFC task, that ON parasols strongly rectify signals unlike OFF parasols (Ala-Laurila, 2014). In the same study, the authors showed and that OFF parasols are more sensitive to dim flashes and flash discrimination is asymmetric between the two pathways. The novelty of the current study is that it provides a contextual link between these response properties and human perception near visual threshold. However, the discussion doesn't sufficiently describe these past findings upon which the current work is based.

Arguments: We agree with the reviewer that the original manuscript did not describe our earlier key findings on the ON and OFF pathway functionality in the primate retina clearly enough and that one of the major findings underlying the novelty of the current paper relates to linking human perception to the updated knowledge of the retinal circuit function at the lowest light levels.

Action points: We have now updated the Introduction (P2, line 17–P3, line 15) by providing a sufficient conceptual link to the previous findings. We have now highlighted more clearly the relation of the current study to the earlier paper (Ala-Laurila & Rieke, 2014), characterizing the

difference in the linearity/nonlinearity of the retinal OFF / ON outputs of the rod bipolar pathway for dim light detection. We have also modified the presentation so that it now clearly highlights the novelty of the current work in linking the retinal ON pathway to human perception for the classical dim light detection task in darkness.

2. Second, the subunit model described here doesn't take into consideration the effects of feedback inhibition (Grimes, 2015) or crossover inhibition (Cafaro, 2013) by amacrine cells other than All, which can regulate gain of signal transmission at RBC synapse. Coupling between rods and cones at dim lights can also provide an alternative high gain pathway for rod signals initiated by single or multi-photon absorption events. What are the relative contributions of these factors to spike generation at these light conditions?

Arguments: Thanks for the comment. The earlier version of the manuscript did not specify clearly enough that the relevant retinal pathway for dim light detection in darkness is the rod bipolar pathway, which constitutes the most sensitive readout of sparse photon responses in the retina. The functional relevance of feedback inhibition and cross-over inhibition for the light detection or light discrimination task is not well established at the visual threshold. Feedback inhibition (Grimes et al., 2015) may constitute an important mechanism for timing related to visual computations, as it sharpens the response properties of single-photon responses as seen by the retinal circuitry downstream of rods. Crossover inhibition, in turn, likely contributes to responses at higher light levels, but its potential role for dim light detection in darkness is not well established. An earlier study (Cafaro et al., 2013) shows that crossover inhibition in ON parasols will be only minimally activated by light increments over a large range of light intensities (0.01 R*/rod/s – 10 000 R*/rod/s). The coupling between rods and cones indeed provides an alternative route for rod-driven signals, but this pathway (known as the secondary rod pathway) is less sensitive than the primary rod-bipolar pathway and does not contribute at the sensitivity limit of vision where our experiments are performed (Grimes et al., 2018). However, we fully agree with the reviewer that we need to highlight those established retinal mechanisms that are essential for the high sensitivity of the rod bipolar pathway.

Action points: We have now added a new paragraph to the Introduction (P2, lines 17– 31) of the paper highlighting that the most sensitive rod signals traverse the retina via the rod bipolar pathway. We further highlight the key mechanisms underlying the high sensitivity of this retinal circuitry to the sparse single-photon responses present in the rod layer at the lowest light levels as well as the key differences in the ON and OFF RGC outputs of this most sensitive rod pathway in the mammalian retina. Finally, we now emphasize in Methods that excitatory synaptic inputs rather than inhibition drive parasol spike responses to light close to the visual threshold and that the model is not making predictions about response timing (P14, lines 4–8): *"Thus, this model is not meant to capture the time course of responses but instead focuses on the impact of thresholding nonlinearities. The model considers a single set of excitatory subunits, as we have shown previously that the excitatory synaptic inputs and the spike outputs of ON parasol RGCs share similar sensitivity and thresholding nonlinearities¹⁰."*

3. Third, the perceptual task presumably depends on retinal output from not just ON parasol pathway, which might be the dominant pathway, but on all retinal pathways that carry information about the stimulus (dim flashes). Without knowing how these other pathways contribute and how the signals from those pathways are processed in downstream circuits in the LGN and V1, it seems premature to conclude the current framework fully describes the mechanisms governing human perception at threshold.

Arguments: The reviewer raises a concern about the potential contribution of different retinal pathways to downstream processing of the weakest light signals. The earlier version of the manuscript did not clearly highlight that it is well established that the dimmest light signals are processed by the most sensitive retinal pathway, the rod bipolar pathway, which has a higher sensitivity than secondary and tertiary rod pathways and all cone driven pathways (Grimes et al. 2018). We have now highlighted that ON and OFF parasol ganglion cells are the best proxy for the most sensitive readout ganglion cell types of the rod bipolar pathway. Their functional and morphological similarity to mouse ON and OFF alpha cells makes them the best candidate primate RGC types for being excellent proxies of the most sensitive ON and OFF outputs of the rod bipolar pathway. In the mouse retina, we have indeed shown (Smeds et al., 2019) that ON and OFF sustained alpha RGCs provide a more sensitive readout of the rod bipolar pathway than any other RGC type. In the primate retina, a full-scale sensitivity classification across all RGC types (> 20 types) has not been done in darkness or at other light levels. However, our pilot data does show that ON and OFF parasol RGCs are more sensitive to the stimuli tested here than for example ON and OFF midget RGCs. This is mainly due to a larger rod pooling in ON and OFF parasols as compared to ON and OFF midget cells due to their larger receptive fields. As the reviewer also suggests, ON and OFF parasols, indeed, are very likely to be the dominant and/or most sensitive readout of the rod bipolar pathway.

Action points: We have now added a new paragraph (P2, lines 17–31) to the Introduction outlining how the dimmest light signals traverse the retina via the rod bipolar pathway, which is the most sensitive rod pathway. We highlight that pooling across thousands of rods in this retinal pathway is a key contributor to its high sensitivity. Furthermore, we now point out that we have recently established in the mouse retina that ON and OFF sustained alpha RGCs are the most sensitive readouts of the rod bipolar pathway across all (> 40) different RGC types in the mouse retina. We now highlight the similarity in response properties in darkness between ON and OFF parasols and mouse alpha RGCs and thereby justify that using parasols as a proxy for the highly sensitive ON and OFF readout RGCs of the rod bipolar pathway in darkness. (P4, lines 15–19).

4. A key findings of the study is that adding a nonlinearity downstream of the retina allows recapitulation of the performance curve (aka, the location and amount of dip) obtained from psychophysical experiments. While exploring this nonlinearity may not be achievable in the current setup, the importance of this nonlinearity cannot be overstated. For example, when rods die in inherited retinal diseases, RGCs can be targeted for stimulation using optogenetics or prosthetics. In the absence of photoreceptor inputs, the ability to detect and discriminate dim lights would inevitably depend on downstream circuits and their nonlinearities. Studied such as

Kremkow et al. 2014, suggest that such nonlinearities may be engaged in ON thalamic pathways but not in OFF thalamic pathways. It would be nice if the authors could discuss these in the context of the present study.

Arguments: We agree with the reviewer about the significance of the nonlinearity downstream of the retina. The reviewer also correctly points out that exploring the properties of this nonlinearity is limited using current techniques. Nonetheless, we agree that it is relevant to understand how the downstream circuits decode retinal signals, including their nonlinearities when considering strategies for addressing visual defects. There is indeed good evidence of nonlinear signal processing in thalamic neural pathways. Thanks for suggesting a good reference (Kremkov et al, 2014), as well.

Action points: We have now expanded the consideration related to nonlinear signal processing in downstream thalamic pathways (P8, lines 12–13). We have added the suggested reference (Kremkov et al., 2014) and another reference (Lee et al., 1983) suggested by the reviewer 1. We agree with both reviewers that these references are good additions to the discussion. We also added the following sentences to the Discussion related to the reviewer’s point about visual restoration: (P6, line 31–P7, line 1): “*This distribution of labor between the retinal ON and OFF pathways in driving cortical responses will be important to consider in attempts to restore visual signals in visually impaired patients.*”

5. From Figure 2 e, f, it appears that OFF parasols, on average, performed slightly better at detection than ON parasols. The threshold intensity for detection is $<10 R^*/RGC$ for all OFF parasols, while $>10 R^*/RGC$ for several ON parasols. This is also observed in Supp Fig 2, by the leftward shift of the 2AFC detection curves for the OFF parasol RGC. These results run contrary to the authors’ claim that both these cell types have similar detection performances. I wonder if the population variance can explain the discrepancy. Is the small dark blue circle the population mean? If it is not, then the authors should include the population mean the SEM. If it is, then the authors should increase the size of the marker and include the SEM for the population. These values should also be included in the figure caption.

Arguments: We appreciate the reviewer’s comment that the SEMs for the detection tasks were missing in Fig. 2e and f for ON and OFF parasols. This made it difficult to assess the similarity between the mean values. The small dark blue circle is indeed the mean, as the reviewer suspected, and we have now added SEMs to clarify the situation. Even though OFF parasols perform slightly better in the detection task, the difference between ON and OFF parasols is not statistically significant. As stated in the main text of the original manuscript (P5, lines 3–6): “*OFF parasol performance was worse than that of ON parasol cells for discrimination ($P=0.0007$, Welch’s t -test, Cohen’s $d = 2.55$) but similar for detection ($P=0.15$, Welch’s t -test; Cohen’s $d=0.41$; blue symbols in Fig. 2 e & f, see also Ala-Laurila and Rieke¹⁰).*”

Action points: We followed the reviewer’s suggestion and added the SEMs for the detection performance for both ON and OFF parasol RGCs in Fig. 2e and f and we now also state the values in the figure caption as the reviewer suggests.

Minor Comments:

6. Study participants are denoted by S1, ..., S5 (pg 9, line 19), flash strengths are denoted by S_1, S_2,... (pg 16, line 11), and supplementary figures are enumerated as S1-S3 (pg 6, line 26). This can be confusing for the readers. I suggest using different letters to denote flash strength and observer identity.

Arguments: Thank you for bringing up this potentially confusing notation.

Action points: We have now changed the notation for study participants (subjects) from S1, S2, etc to O1, O2, etc. (see P9, lines 5–6; P11, line 7; P15, lines 12–13) and Fig. 3c.

7. On page 5, the authors test the extent to which difference between psychophysical responses and ON parasol responses depend on number of activated RGCs. The authors do this by increasing the number of subunits from 3 to 6. What is the reason for doubling the number of subunits? Does the size difference between the stimulus spot projected at 30deg eccentricity and the receptive field correspond to 3 additional subunits? This needs explanation.

Arguments: The reviewer is correct: the reason for increasing the number of subunits from 3 to 6 relates to the area of the stimulus used in the psychophysics experiments being ~2 times larger than the RF of a single RGC.

Action points: We have now clearly indicated the reason why the subunit number is twice larger for human psychophysics, as compared to RGCs and made changes to the text in P6, lines 10–18: *“Next, we tested if access to more than a single RGC could explain the difference between behavior and ON parasol responses, since the retinal stimulus size in psychophysics corresponds to ~2 times the RGC receptive field. Increasing the number of subunits, as demonstrated by doubling their number from 3 to 6 (M1, $n = 6$, see Fig. 3e) only shifted the curve upward in M1, but did not increase the dip. However, adding the second nonlinearity as in M2, resulted in a good fit to the human behavioral data (Fig. 3e; M1, $n = 6$, $\theta_1 = 2$, $\theta_2 = 2$; for model robustness, see Supplementary Fig. 4). This result is consistent with additional downstream thresholding of ON parasol signals in the brain at the sensitivity limit of vision.”*

8. Nonlinear processing of visual signals in contrast discrimination tasks has been shown to produce dipper-like curves (Legge & Foley, 1980; Nachmias & Sansbury, 1974). The authors should consider citing these studies. The authors should also cite references for the subunit model (Hichstein, 1976; Shapley, 1979; Crook, 2008).

Arguments: Thanks for a good suggestion. We have now added the suggested references.

Action points: We added the suggested references (Legge & Foley, 1980; Nachmias & Sansbury, 1974; Hochstein, 1976; Shapley, 1979; Crook, 2008) in the text (P4, lines 1–5): *“Such dips have been reported in contrast discrimination tasks for retinal ganglion cells¹⁷, and their*

mechanistic origin could be understood in the context of nonlinear subunits in their receptive fields^{18–22}. Similarly, dips have been previously reported for human psychophysics in various stimulus conditions^{23–25}.

9. The authors should cite the study by Soto et al., 2020, which shows evidence for conserved midget and parasol pathways in human retina.

Arguments: The reviewer raises a good point that further justifies our approach, where we use non-human primate retinas as a model system for the human retina related to ON and OFF parasol signals. This reference (Soto et al., 2020) is a good reference for this justification.

Action points: We have now added a reference to Soto et al. (2020) as justification for using an RGC model based on recordings from non-human primates to predict human psychophysics. See P8, lines 22–25: *“Primate retinas were used as a proxy for human retinas, since harvesting their dark-adapted ex vivo retinas at the uttermost sensitive conditions is more feasible than human retinas. It has also been recently shown that the spatiotemporal response properties of human and non-human primate ON and OFF parasol cells are very similar³²”*

10. What are the parameters for high vs low noise and high vs low threshold in Supplementary Fig. 3? Is the noise regime set by the quantum efficiency of photon absorption?

Arguments: Thanks for highlighting that this information was missing. We have now added the information about the model parameters and their units. The reviewer is indeed correct: The noise regime is defined in the isomerization (R^*) regime and takes into account the quantum efficiency of photon absorptions.

Action points: We have updated Supplementary Fig. 3 for clarity and added the parameter values to the figure legend.

11. The authors state that “All observers had normal uncorrected vision” (pg 7, line 24). Were Visual Field Test or Perimetry Test done to confirm that the subjects had normal vision?

Arguments: Thanks for pointing this out. We had omitted information about the visual tests done on test subjects in the original manuscript.

Action points: Now we have added specifics about the visual acuity and visual field tests in the psychophysics section of the Methods (see P9, lines 8–12): *“This was validated by an ophthalmologist testing all subjects for visual acuity and visual field sensitivity. All observers had normal monocular visual acuity (range 20/25 – 20/16) and no significant visual field defects within the eccentricity (<20°) were observed in a perimetry visual field test (Octopus 900, Haag-Streit Diagnostics, Switzerland).”*

12. The statement “our results indicate that response to individual photons interact, providing evidence for the perceptual relevance of detection of coincidence photons” is confusing because it does not answer the question whether humans can perceive single photons. In fact

this paragraph reads that single photons are absorbed by rods but downstream processing makes only (or largely) information about coincident photons available to the brain for perception. Also, “humans can see a photon” phrase can be misconstrued. Seeing is an experience that is shaped by factors unrelated to the visual task. I think this paragraph should be revised to make the interpretation clear.

Arguments: We understand the concerns that the reviewer has about the clarity of this discussion section, particularly related to the phrases “*individual photons interact*” and “*seeing a single photon*”.

Action points: We have now revised the Discussion section (see P7, line 7–P8, line 2). We have replaced the aforementioned unclear statements with more accurate ones. More importantly and related to the comments by reviewer 3 (major comment 2): We have now expanded the discussion about the argument “*humans can see a single photon*” and discuss it in relation to the other paper studying single-photon detection on human psychophysics (Tinsley et al., 2016).

Reviewer #3: Overview

The authors aim to answer the question of whether human visual processing at extremely low light levels is linear, or whether there is a nonlinear threshold in the retina before visual signals reach the brain. This manuscript presents new evidence that may distinguish between these two alternatives. The primary approach is to use measured performance in detection vs. discrimination tasks to distinguish between the models, which make different predictions for performance in these two types of tasks. In particular, the authors report two experiments: an in vitro study using output signals of the primate retina as a proxy for human retinal outputs, and a two-alternative forced-choice (2AFC) behavioral study with human subjects. The authors conclude that both experiments support their hypothesis that there is a nonlinear threshold in the human retina, and that there is additional thresholding in the brain.

This is an interesting, timely, and well-conceived study. The question of whether there is a nonlinear threshold in the retina at extremely low light levels is important. In particular, it is relevant to the question of whether humans can detect single photons, which has been a subject of debate essentially since photons were discovered, and has implications for possible future quantum science experiments. The hypothesis that there may be a photon coincidence requirement (nonlinear threshold) somewhere in retinal processing, and that single-photon detection is therefore not possible, has been an important aspect of this debate. Evidence in support of this hypothesis therefore has a major impact on the study of human vision at the single-photon level. The methods used in this article are appropriate and provide new and useful experimental data for answering the main question. In most cases the data support the conclusions, although some conclusions may be overstated (see detailed notes).

We appreciate the overview of our study, particularly related to its importance and impact in the field. We also note with gratitude the reviewer’s positive comments on the methodology and the implications in the field of quantum science (see also the attached manuscript Tiihonen et al.

related to this field). We appreciate the reviewer’s critique related to the presentation style of some of the results. As outlined in detail below, we have now carefully recrafted the tone of the revised manuscript in multiple places to address this concern.

Major comments:

1. There are two main issues that should be addressed. First, on page 2, the authors mention that recent results from primate retinas and mice have suggested that the linear model of visual processing may be incorrect. It seems that the primary goal of the current work is to find out whether these results also apply to humans—an important question! The manuscript should include a statement along the lines of “A study that combined retinal signal measurements with behavioral studies in mice found X, and measurements with primate retinas suggested that something similar is also happening in primates. The mouse experiment can’t be replicated with human subjects, so in order to know whether this same effect happens with humans, we need to know Y and Z. Our measurements with primate retinas provide Y and our human behavioral experiments provide Z, therefore we can conclude that what was observed in mice is also likely happening in humans.” Without this explicit context, I believe it will be difficult for a typical reader to understand whether these results really prove the authors’ broader hypothesis.

Arguments: Thanks for kindly pointing out on how to improve the presentation to communicate the rationale of the study better to a typical reader. Our original Introduction section did not clearly describe our own findings on retinal circuit mechanisms in primates and the linkage we manage to do on mouse models between retinal circuit function and behavioral perception. A similar well-pointed observation was made by the reviewer 2 (see above). We agree that it is important to clarify why the mouse experiments cannot be replicated in humans, where transgenic approaches such as in mice are not feasible. We also believe that it is important to highlight better that the non-human primate retinas are a good model system for the human retinas particularly at these extreme recording conditions at the visual threshold, where the retinal preparation conditions need to be of the highest quality feasible.

Action points: We have now revised the Introduction to address the points that the reviewer raises and help an average reader to understand the overall logic of the paper and how the key hypothesis of the paper is related to previous findings. We also now better explain why a different approach now is needed as compared to those used on mice and described in the Smeds et al. 2019 paper (see P2 lines 14–16). Furthermore, we justify now better why ON and OFF parasols are good proxies for the high-sensitivity readouts of the rod bipolar pathway such as alpha RGCs in mice (see P4, lines 15–19) and include a more extended justification for using primate retinas as a proxy for the human retinas at the most sensitive conditions (P8, lines 22–25).

2. Second, the authors must directly address how their results are or are not consistent with behavioral evidence for single-photon detection by humans, in particular Tinsley et al. Tinsley et al claimed to have evidence for single-photon vision in a 2AFC task, so this manuscript’s claim that single-photon vision is impossible is in direct conflict with that result. Observing an effect

tends to be more convincing than arguing why it is impossible. It is not necessarily these authors' responsibility to explain why the Tinsley study found different results—and there are many possibilities; for example, it had low statistical power and didn't have a control group—although if they have insight, it would certainly add to the scientific discourse on the subject. But if they are going to make the case that the “decades-old question about whether humans can see a single photon” has now been answered, it requires more discussion to put that conclusion in context, especially in light of contradictory evidence. The disagreement with other recent studies, e.g. Tinsley et al, is one reason why this research is significant.

Arguments: The reviewer raises an excellent point regarding the discrepancy related to “*the decades-old question about whether humans can see a single photon*” between our results and those reported by Tinsley et al. (2016). We did not address this discrepancy in the original manuscript since we have been working extensively on another project with Prof. Anton Zeilinger (Institute for Quantum Optics and Quantum Information, Vienna, Austria) and his group using a single-photon source. Our project included single-photon detection experiments from the retina to perception and a replication of the psychophysics experiments performed by Tinsley et al. (2016). In addition, we have carried out experiments using the single-photon light source on isolated rod photoreceptors. Our project directly links the retinal processing of single-photons across the rod bipolar pathway to human psychophysics experiments. The reviewer points out “*it is not necessarily these authors' responsibility to explain why the Tinsley study found different results - and there are many possibilities.*” We have taken this responsibility seriously and have carried out an extensive set of experiments. However, this project represents a substantial and fundamental dataset from rods to psychophysics relying on a single-photon light source and we believe that presenting these results deserves its own paper with its own focus. We now append to this submission an early draft of this manuscript containing its abstract and main figures with legends (Tiihonen et al., in preparation). We cite the key findings of this manuscript to the extent that they are relevant in addressing the reviewer's question. In “Action Points” below we describe how we have modified the Discussion section of the revised manuscript to take into account our findings presented in this unpublished study by Tiihonen et al.

Tinsley et al. (2016) (taking into account both the main paper and its supplement) carry out both classical psychophysics experiments and single-photon detection experiments in the 2AFC framework. Classical psychometric functions and single-photon experiments probe the same visual perception system but do that in two fundamentally different stimulus intensity regimes. Classical psychometric functions quantify how detection performance changes from chance level (50 % in the 2AFC task) to “*always seen*”, and the intensity regime where this transition happens is typically probed using a standard Poisson light source. Single-photon detection experiments, on the other hand, aim to test if a single photon produced by a single-photon light source or alternatively by an extremely dim standard Poisson light source can cause a statistically significant detection performance difference as compared to “no photon” trials. Classical psychometric functions are therefore measured in a stimulus intensity regime, where the detection performance starts to deviate systematically from the chance level, whereas single-photon detection experiments occur at stimulus intensities where the retina has access to

maximally just one photon at a time and detection performance is near chance. The single-photon regime thus requires a huge number of repeats to test if the response distributions between “no flash” and “flash” trials deviate from each other.

The exact number of repetitions needed in the single-photon regime is constrained by the minimal losses of photons from the retina to perception. This minimum total loss of photons impacts the psychometric functions measured in classical detection experiments. Thus, the photon loss model has to be consistent with the psychometric function, as the same neural circuits process both stimuli. This photon loss model should be a starting point for any single-photon detection experimental design, irrespective of whether it is performed using a Poisson source or a single-photon source. However, Tinsley et. (2016) never relate the minimum photon losses that are present in the visual system in their classical psychometric functions to the trial numbers that are minimally needed for single-photon detection experiments presented in their paper and its supplement. Thus, the fundamental challenge related to the Tinsley et al. (2016) study is that they never relate their two datasets (psychometric functions and single-photon experiments), nor do they require that the datasets can be interpreted within a single theoretical framework. In their case, the upper theoretical limit on the performance of an ideal single-photon detector (see dashed line in Fig. 3c in Tiihonen et al. manuscript) is completely out of bounds with what would be expected from a psychometric function matching their and our data on the 2AFC detection task. In their paper, they only show the theoretical limit in their Fig. 2a, but it is not pointed out and/or noticed that this loss model is not at all consistent with the psychometric functions matching their data (their Supplementary Fig. 3). In fact, the loss model underlying their theoretical limit predicts 100% detection performance for an intensity ($\sim 3 R^*/\text{flash}$ corresponding to 20 photons present at the cornea, see Fig 3c in Tiihonen et al. manuscript) where they and we measure performance to still be at the chance level. This light intensity is in fact ~ 100 times higher than the intensity used for single-photon detection experiments.

The key question then becomes, taking into account the realistic loss model that predicts both their and our 2AFC detection experiments, what is the expected number of trials needed for statistically significant results in a single-photon detection experiment? Tiihonen et al. (see attached MS) now computes the expected number of trials taking into account a loss model that is consistent both with the psychophysical classical detection experiments as well as retinal output signals and shows that one would need millions of repeats. This means that it is totally unfeasible to reach a statistically significant proof of detection of a single photon within realistic experimental time frames. To verify this, we performed the same experiments as Tinsley et al. (2016). We were able to increase the efficiency of the experimental protocol crucially by utilizing a timing module integrated to our single-photon source. This timing module allowed us to evaluate in real time, whether a heralded photon pair was generated or not. On the other hand, Tinsley et al. (2016) identified only post-hoc the trials when a photon pair was created. Thus, their experiment contained a lot of blank-blank trials and they estimated that out of 30 000 trials only 2420 contained a photon. On the other hand, our source was operated in an optimized way, which allowed us to detect blank trials in real time, and allowed us to use a quick succession of repeated laser pulses on the go, until a photon pair was created. In our experiment, the flash duration was 5 ms and the pulse frequency of the laser was 20 kHz. This

allowed us to generate a total of 100 pulses in 5 ms. In this experimental realm, we were continuing pumping laser pulses until a photon pair was created. The 5-ms window worked fine for us and we did not have to show the large number of “blank” - “blank” trials such as Tinsley et al. 2016. Instead, we were able to create exactly one photon vs. blank each time that a subject was tested on a single trial. Our 2695 total trials are to be compared to those 2400 of Tinsley’s post-hoc identified trials (see Tiihonen et al. attached manuscript draft). So, the total number of trials was similar in our case. However, we did not find any effect that was statistically different from the chance level, despite that our observers having almost identical psychometric functions in the classical detection experiment as the ones in Tinsley et al. (2016). These results are reported in Tiihonen et al. manuscript. (attached) and the early findings are also reported in an ARVO abstract (Tiihonen et al., 2020).

Why then do Tinsley et al. (2016) claim that single-photon detection experiments produce statistically significant effects? We believe that the strong claim made in their paper - humans can detect the absorption of a single photon in a single rod - requires a strong statistical proof (see also the original published reviewer response of Tinsley et al. (2016) paper). Their proof for single-photon detection relies on barely significant p-values for two extremely dim stimulus intensities (0.2 and 1.0 photons at the cornea; these data points by Tinsley et al. (2016) are highlighted by arrows in Tiihonen et al. manuscript, Fig. 3c): $p = 0.0545$ (Fisher exact test, one-tailed) for the single-photon source and $p = 0.036$ for the 5 times more intense Poisson source (Fisher exact test, one-tailed); by pooling these two with Fisher’s method they got $p = 0.014$ (Tinsley et al. (2016), main article Fig. 2A and their figure R1.1). We repeated this very experiment (Tiihonen et al., attached MS, Fig. 3d) using larger trial numbers and show that there is no statistical significance for single-photon detection in exactly the same conditions as Tinsley et al. (2016) used. We also observe change performance for flashes that are 20-fold brighter; at these flash strengths the model that Tinsley et al. (2016) used in single-photon experiments (see Fig. 3c dashed line, Tiihonen et al. attached manuscript) predicts near-perfect detection performance.

There are also other anomalies in the statistical analysis of the Tinsley et al. (2016) study, which we cannot replicate: One of their arguments supporting single-photon detection is presented in their reviewer response Fig. R1.1 (replotted below in Fig. 2). This figure highlights results for the subset of responses for the two stimulus intensities that subjects reported as high-confidence. By pooling data from both stimulus intensities, Tinsley et al. (2016) obtained a p-value of 0.0003, thus claiming that the deviation from the chance level would be statistically significant. However, they get surprisingly high significance ($p = 0.001$, Fisher exact test, one-tailed) for the 5 times lower intensity, whereas the higher intensity gives $p = 0.029$ (Fisher exact test, one-tailed). Taken into account the intensity difference and the two p-values, the realization of such a result is highly unexpected. We repeated the same experiment using the same intensities and using larger trial numbers (Tiihonen et al., attached manuscript, Supplementary Fig. 1f). We observe no statistical significance for single-photon detection in these exact same experimental conditions.

In summary, we propose that papers presenting classical detection experiment data and single-photon detection data should 1) provide a unified theoretical framework for the loss model explaining the two datasets, 2) compute the expected trial numbers needed for the single-photon detection experiment and 3) repeat the Tinsley et al. (2016) experiment with larger or similar trial numbers and compare the results. We have done all of that in Tiihonen et al. (see attached MS) and conclude that humans cannot detect single photons.

Fig. 2. Reprinted from Tinsley et al. (2016) reviewer response (Fig. R1.1). Original legend: “Probability of correct response in quantum light source single-photon experiments (original data, 3 subjects) and Poisson light source with an average photon number one (new data for 4 subjects). The probability of correct response is shown for combined ratings (brown bar) and the high-confidence R3 ratings (green bar) together with corresponding p values as well as the Fisher combined p values.”

Action points: We have now followed the reviewer’s suggestion and added a discussion about the discrepancy between our results and those by Tinsley et al. (2016) paper (P7, lines 7–19 in the revised Discussion section). We hope that this revision helps the reader to assess the differences between our results and those obtained by Tinsley et al. (2016).

3. Finally, the statistical analysis is missing a discussion of uncertainty in the measured values, particularly the threshold θ , including sources of uncertainty from the light intensity conversions. A best fit value of θ was determined by fitting a model, but would other values of θ also be consistent with the experimental data within uncertainty? Some plots showing experimental data also lack error bars.

Arguments: Thank you for pointing out that we need a better robustness analysis of our model. The original manuscript showed the effects of changing every parameter of the model (Supplementary Figure 3) but did not compare the change against the psychophysics data nor the fitted model (M2). It only gave a qualitative understanding of the impact of each model parameter to the model fits. We have now added a new robustness analysis (see Action points below) and also taken into account the original lack of error bars (see below). In particular, related to the reviewer’s question about the threshold parameter θ : The robustness analysis shows that doubling or halving the threshold θ completely misaligns the model with the

observed data (see Supplementary Figure 4 and Action points below). The light intensity conversions from photons to R*s are constrained as tightly as possible with the best possible measurements available for monkey and human rods as explained in Methods.

Action points: We have now added a robustness analysis for each model parameter in a new Supplemental Figure 4. This figure shows how the model performance changes in comparison to the psychophysics data and the fitted model when one model parameter at a time is doubled and/or halved. In addition, we have added SEMs to Figure 2e and f, as also requested by the reviewer 2. We also added the following sentences to the Methods section of the paper (starting on P14, line 24): “*Supplementary Fig. 4 shows model robustness when changing each key model parameter either by doubling or halving their values. The main conclusions of the paper – namely that the RGC dataset and psychophysical datasets can be bridged within the “new model” (M2) and not within the “classical” model (linear retina model) – are robust across any reasonable parameter space.*”

Minor Comments:

4. Page 2, Line 8: I disagree somewhat with the characterization of the two models as “classical” and “new” – I think neither model was previously totally accepted or supported by evidence, which is in fact a justification of the importance of the current study. **However, I may be wrong.** My suggestion would be to call them “linear” and “nonlinear” models.

Arguments: We understand the reviewer’s point about the naming conventions of the models. We were also considering the reviewer’s suggestion to call the models “*linear*” and “*nonlinear*”. The challenge is that both models have a nonlinearity in the brain and the main difference being that the retinal part of the model underlying increment detection is linear in one of the models (“classical model”) and nonlinear in the other (“new model”). Thus, we ended up calling the models “classical” and “new”. However, we define the linear and nonlinear model elements in the model descriptions. We hope that this terminology is acceptable.

5. Page 2, Line 13: Suggest briefly introducing ON and OFF pathways for the reader – this is not a specialized publication in psychophysics/vision science.

Arguments: We appreciate the reviewer’s point that more background information is necessary to ensure that the average reader understands our study and its results. We fully agree with the reviewer and have followed this suggestion in the revised manuscript.

Action points: We have now added a new paragraph to the introduction where the ON and OFF pathways are introduced, as also requested by reviewer 1. We have now added a new paragraph to the Introduction (P2, lines 17–31) of the manuscript where we highlight that signals originating from single-photon responses at the lowest light levels are mediated by the rod bipolar pathway. We briefly explain how the signals are processed along the ON and OFF branch of this pathway to ON and OFF RGC outputs.

6. Page 2, Line 20: “in which perception [near threshold]” relies on retinal outputs ...” Or is it really perception at all light levels?

Action points: We have now rewritten the sentence to clarify that we indeed mean perception near the visual threshold. The sentence (P3, lines 1–5) now states: “*These recent findings challenge the classic model for perception at visual threshold, and suggest a new model in which perception relies on retinal outputs originating in the ON pathway, and those signals are shaped by a thresholding nonlinearity that limits access to individual single-photon responses (see Fig. 1b).*”

7. Figure 1 a-b: Why is there an arrow directly from spikes to behavioral output in 1b? I would think all behavioral output needs to involve the brain?

Arguments: We appreciate that the reviewer highlighted that it is not clear what the purpose of the arrow from spikes to behavioral output signifies in Fig. 1b. Originally, we intended this arrow and the corresponding one in Fig. 1a to denote that classical models of psychometric functions connected photon distributions to perceptual outputs (arrows) without having constraints related to RGC outputs. However, we understand that this utilization of arrows in the figure was not explained well enough. In addition, the argument does not seem very important as we now re-evaluate the original presentation.

Action points: We removed the confusing arrows in Fig. 1a and b that went from photons and spikes, respectively, to the behavioral output.

8. Figure 1 Caption Line 4 – Typo “refence”.

Action points: It is now fixed.

9. Figure 1 c-d: It would be helpful to give the values of N, S1, S2, and S3 in this figure (it's possible to tell from the plots, but the values would still be nice). Also, if the conclusion is that the threshold is 2 photons, why does this figure show an example where the threshold appears to be 6?

Arguments: We agree with the reviewer that the values of N, S1, S2 and S3 would be helpful and have now taken this into account (see below). We also understand the reviewer's point that it was hard to follow the logic of our original presentation. The primary purpose of Fig. 1 is simply to demonstrate the effect of linear and nonlinear signal processing without strictly connecting the mean intensity values (R^*) to subunit threshold values. We believe that it is easier to see how the location of the “dip” in ΔI_{JND} function (panel h) correlates with the location of the nonlinear threshold with such a presentation: The noise level in this schematic is $2 R^*$, meaning you need 4 additional R^* s to get over the threshold of 6. Thus, the dip occurs at a reference intensity (I_{ref}) of roughly $4 R^*$. However, we did not choose the threshold values randomly in this demonstration, and it is easy to notice later in the paper that the ΔI_{JND} functions in this demonstration closely resemble those measured in OFF and ON RGCs. The reason for

the mismatch between the threshold of $6 R^*$ in Fig. 1 and the subunit threshold value of $2 R^*$ is that each RGC collects signals from multiple subunits (three according to our model). Thus, the effective threshold at the RGC level (as R^* s refer now to R^* s per RGC) will be higher than at the subunit level (R^* s per subunit) and this is why the dip appears at higher reference intensities than $2 R^*/\text{RGC}$ in Fig. 2f. However, we have tried to avoid the complication of having to address this in the context of Fig. 1. We believe it is simpler to understand the logic in Fig. 1 without having to think of a more complex scenario where both subunits and nonlinearities have to be considered.

Action points: We have now added the distribution means ($N=2$, $N+S_1=6$, $N+S_2=10$, $N+S_3=14$) to the figure legend, as well as an explanation of why the dip occurs at roughly $4 R^*$. In addition, we now emphasize in the main text that Fig. 1 is a schematic illustration of how linear and nonlinear models perform on detection and discrimination tasks (P3, line 20).

10. Figure 1 The combination of linear and nonlinear scales makes it hard to tell where the “dip” occurs and how it relates to the threshold.

Arguments: Thanks for highlighting this potential source of confusion. We clarify in the previous answer how the “dip” relates to the threshold and implement changes to the paper as outlined in the Action points below.

Action points: We now clarify the relation of the dip to the threshold in the legend of Fig. 1. We added the following sentence at the end of the Fig. 1 legend (P21, lines 15–17): “The dip in h occurs approximately at a reference intensity of $4 R^*$, as 4 additional R^* s are needed to surpass a threshold of $6 R^*$ when the noise level is $2 R^*$.”

11. Page 5, Line 3 – Additive and multiplicative noise levels were fit as model parameters, but I can’t find the actual noise values in the manuscript. Were reasonable noise levels required to achieve the best agreement with data?

Arguments: Thanks for pointing out that the actual values of the noise parameters of the model should be more clearly stated in the manuscript. We presented the other model parameters already in the original manuscript, but the noise parameters were missing. This has now been fixed. However, the noise parameter values can be tricky to interpret as it is not easy to intuitively grasp the multiplicative noise in particular. We therefore plotted the Fano factor for the RGC model output and compared it to the observed Fano factor for the ON RGC spike output compared it to the observer Fano factor for the ON RGC spike output (mean \pm std; see Fig. 3). This comparison illustrates that the used model parameters result in a Fano factor that matches the shape and magnitude of the empirically observed Fano factors for the ON RGCs.

Fig. 3. Fano factor for the RGC model outputs and the ON RGC spike outputs (mean \pm std) as a function of stimulus intensity.

Action points: We added the following sentence at the end of Fig. 3 legend: “For the model robustness analysis and parametrization, see Supplementary Figs. 3 & 4.”. Supplementary figures give a complete model parametrization and its robustness analysis for those readers who want to understand the model performance in greater detail.

12. Page 5, Line 18 – I believe the language here is somewhat more conclusive than is justified by the evidence presented. This work shows that a plausible model is in good agreement with experimental data, which is a very reasonable way to approach the problem, but is not quite direct evidence comparable to the mouse study in Reference 10. Instead of “human behavior at absolute threshold relies on retinal output signals provided by ON RGCs”, I would suggest “our results support the hypothesis that human behavior at absolute threshold relies on retinal output signals provided by ON RGCs”. Similarly, say “This result shows that retinal ON and OFF pathways likely carry out distinct functional roles ...” As discussed in the introductory comments to this referee report, a clearer explanation of how this work fills in the gaps of previous primate and mouse studies would help the reader judge the strength of the conclusions. The authors should also discuss alternative explanations of their results or other models that could explain the data, if there are any.

Arguments: We agree with the reviewer about the suggested edits to the tone of the presentation.

Action points: We implemented the reviewer’s suggestion. Now we state (P6, starting on line 23): “First, similar to previous results in mice¹⁰, our results support the hypothesis that human behavior at the absolute threshold relies on retinal output signals provided by ON RGCs. This is true even when the sensitivity of OFF RGC responses is comparable to that of ON RGC responses. This suggests that, at least at the visual threshold, increases in retinal firing rates (e.g. ON RGC responses to light increments) are more effective than decreases (OFF RGC responses to increments) in eliciting cortical responses and behavior.” We have also revised the

Introduction of the paper to provide a clearer explanation of how our current work fills the gaps in previous mouse and primate studies.

13. Page 5, Line 26: “Our results answer the decades-old question about whether humans can see a single photon” – with a setup like that, the reader is expecting a clear “yes” or “no” in the rest of the sentence. Also, as discussed in introductory comments to this referee report, there must be a direct discussion of how these results are/are not in conflict with Tinsley et al.

Arguments: Thanks for bringing this up. We believe that this question is now addressed in its full extent while we address the reviewer’s major comment 2.

Action points: See the response to the reviewer’s major point 2 above.

14. Page 6, Line 3: Suggest changing to something along the lines of “Thus, seeing single photons has likely not been the central optimization goal during evolution.” I don’t think there is any direct evidence for “evolutionary optimization goals” in this study.

Arguments: Thanks for pointing out that this statement should be softened slightly.

Action points: We included the reviewer's suggestion and the sentence now reads (starting on P7, line 29): “Thus, visual perception of single photons has likely not been the central optimization goal during evolution.”

15. Page 9, line 17 – If the value of the ocular media factor is important to the conclusions drawn (for example, the value of the threshold) the authors should note that it is very approximate and known to vary between individuals. (Reference 8 states: “All of these factors vary considerably between individuals, so any calculations we make with them will be approximate.”) This uncertainty has been a major limitation of studies of single-photon vision using classical light sources. If the exact value of the ocular media factor is not important to the conclusions, please say so. If it is, the authors should estimate how it contributes uncertainty to their estimate of the threshold.

Arguments: The reviewer is correct that there can be differences across individuals in factors impacting detection performance (including the ocular media factor). Variation in the ocular media factor would cause lateral shifts of the 2AFC detection curves without changing the shape of these functions. The lateral shifts would directly translate to changes in the 2AFC thresholds that are shown for the 5 test subjects in Fig. 3c (see the blue datapoints). The range for the threshold values on the detection task is: 13.5 - 27.6 R*, mean \pm SEM: 18.6 ± 2.7 R*. Assuming that all of the variation is due to changes in the ocular media, we can conclude that the maximal variation is two-fold within our test subjects. Within this range there is effectively no impact on the estimated model threshold parameter. This is not surprising, since the threshold parameters directly impact the shape of the psychometric functions (the steepness), in contrast the ocular media factor does not impact its shape but causes only lateral shifts (see Field et al. 2005, review). So, there is not much interplay between these two parameters, whereas

releasing both noise parameters, ocular media and threshold parameters allows more interplay between different parameter realizations of the model fits to detection functions. The review by Field et al. 2005 focuses on these challenges in the context of detection experiments alone and their inspections in the “classical model” framework. The major advance of our paper in comparison with earlier contributions is that we now include both detection and discrimination experiments to study if the RGC response properties and psychophysical performance requires the introduction of a retinal nonlinearity. Related to these two experimental paradigms, the key point is that the dip in ΔI_{JND} functions (requiring discrimination experiments in addition to detection experiments) does not occur at all in the linear retina model even if the noise parameters are allowed to change across a wide range of values. Our model robustness analysis (Supplementary Figs. 3 & 4) shows how our model behaves across parameter changes. It should be noted that the main point of the paper is not the exact parametrization of the “new model” but that the data can be explained by that model framework rather than the classical model and that the new model also needs the higher-order nonlinearity to explain quantitatively the bridging of the RGC and the psychophysical datasets.

Action points: We have now added the following sentence (P14, lines 25–28) to the paper: “*The main conclusions of the paper – namely that the RGC dataset and psychophysical datasets can be bridged within the “new model” (M2) and not within the “classical” model (linear retina model) – are robust across any reasonable parameter space.*”

16. Page 13, Statistical Analysis – Include discussion of uncertainty in the measured values, particularly the threshold θ .

Arguments: Thanks again for highlighting the need for a robustness analysis for the model parameters. We believe this comment is related to the reviewer’s major comment 3, to which we elaborate on a robustness analysis of each model parameter, including the threshold (θ).

Action points: See the response to the reviewer’s major point 3 above.

References:

- Ala-Laurila, P., Rieke, F., 2014. Coincidence Detection of Single-Photon Responses in the Inner Retina at the Sensitivity Limit of Vision. *Current Biology* 24, 2888–2898. <https://doi.org/10.1016/j.cub.2014.10.028>
- Cafaro, J., Rieke, F., 2013. Regulation of Spatial Selectivity by Crossover Inhibition. *J. Neurosci.* 33, 6310–6320. <https://doi.org/10.1523/JNEUROSCI.4964-12.2013>
- Crook, J.D., Peterson, B.B., Packer, O.S., Robinson, F.R., Troy, J.B., Dacey, D.M., 2008. Y-Cell Receptive Field and Collicular Projection of Parasol Ganglion Cells in Macaque Monkey Retina. *J. Neurosci.* 28, 11277–11291. <https://doi.org/10.1523/JNEUROSCI.2982-08.2008>

- Field, G.D., Sampath, A.P., Rieke, F., 2005. RETINAL PROCESSING NEAR ABSOLUTE THRESHOLD: From Behavior to Mechanism. *Annual Review of Physiology* 67, 491–514. <https://doi.org/10.1146/annurev.physiol.67.031103.151256>
- Field, G. D., Uzzell, V., Chichilnisky, E. J., & Rieke, F., 2019. Temporal resolution of single-photon responses in primate rod photoreceptors and limits imposed by cellular noise. *Journal of Neurophysiology*, 121(1), 255–268. <https://doi.org/10.1152/jn.00683.2018>
- Grimes, W. N., Zhang, J., Tian, H., Graydon, C.W., Hoon, M., Rieke, F., Diamond, J.S., 2015. Complex inhibitory microcircuitry regulates retinal signaling near visual threshold. *Journal of Neurophysiology* 114, 341–353. <https://doi.org/10.1152/jn.00017.2015>
- Grimes, W. N., Baudin, J., Azevedo, A. W., and Rieke, F., 2018. Range, routing and kinetics of rod signaling in primate retina. *Elife* 7, e38281. <https://doi.org/10.7554/eLife.38281>
- Hochstein, S., Shapley, R.M., 1976. Linear and nonlinear spatial subunits in Y cat retinal ganglion cells. *The Journal of Physiology* 262, 265–284. <https://doi.org/10.1113/jphysiol.1976.sp011595>
- Kremkow, J., Jin, J., Komban, S.J., Wang, Y., Lashgari, R., Li, X., Jansen, M., Zaidi, Q., Alonso, J.-M., 2014. Neuronal nonlinearity explains greater visual spatial resolution for darks than lights. *Proceedings of the National Academy of Sciences* 111, 3170–3175. <https://doi.org/10.1073/pnas.1310442111>
- Lee, B.B., Virsu, V., Creutzfeldt, O.D., 1983. Linear signal transmission from prepotentials to cells in the macaque lateral geniculate nucleus. *Exp Brain Res* 52, 50–56. <https://doi.org/10.1007/BF00237148>
- Legge, G.E., Foley, J.M., 1980. Contrast masking in human vision. *J Opt Soc Am* 70, 1458–1471. <https://doi.org/10.1364/josa.70.001458>
- Nachmias, J., Sansbury, R.V., 1974. Letter: Grating contrast: discrimination may be better than detection. *Vision Res* 14, 1039–1042. [https://doi.org/10.1016/0042-6989\(74\)90175-8](https://doi.org/10.1016/0042-6989(74)90175-8)
- Shapley, R.M., Victor, J.D., 1979. Nonlinear spatial summation and the contrast gain control of cat retinal ganglion cells. *The Journal of Physiology* 290, 141–161. <https://doi.org/10.1113/jphysiol.1979.sp012765>
- Short, A.D., 1966. Decremental and incremental visual thresholds. *The Journal of Physiology* 185, 646–654. <https://doi.org/10.1113/jphysiol.1966.sp008007>
- Smeds, L., Takeshita, D., Turunen, T., Tiihonen, J., Westö, J., Martyniuk, N., Seppänen, A., Ala-Laurila, P., 2019. Paradoxical Rules of Spike Train Decoding Revealed at the Sensitivity Limit of Vision. *Neuron* 104, 576–587.e11. <https://doi.org/10.1016/j.neuron.2019.08.005>

- Soto, F., Hsiang, J.-C., Rajagopal, R., Piggott, K., Harocopos, G.J., Couch, S.M., Custer, P., Morgan, J.L., Kerschensteiner, D., 2020. Efficient Coding by Midget and Parasol Ganglion Cells in the Human Retina. *Neuron* 107, 656-666.e5.
<https://doi.org/10.1016/j.neuron.2020.05.030>
- Tiihonen, J., Dovzhik, K., Tavala, A., Zeilinger, A., Ala-Laurila, P., (attached unpublished manuscript). Retinal loss of single photons sets a fundamental limit to quantum-resolution visual perception.
- Tiihonen, J., Tavala, A., Dovzhik, K., Ala-Laurila, P., 2020. Retinal loss sets a fundamental limit to the variance of visual signals originating from single-photon stimulation. *Investigative Ophthalmology & Visual Science* 61, 1106.
- Tinsley, J.N., Molodtsov, M.I., Prevedel, R., Wartmann, D., Espigulé-Pons, J., Lauwers, M., Vaziri, A., 2016. Direct detection of a single photon by humans. *Nat Commun* 7, 12172.
<https://doi.org/10.1038/ncomms12172>
- Wassle, H., Boycott, B.B., 1991. Functional architecture of the mammalian retina. *Physiological Reviews* 71, 447–480. <https://doi.org/10.1152/physrev.1991.71.2.447>
- Westö, J., Martyniuk, N., Koskela, S., Turunen, T., Pentikäinen, S., Ala-Laurila, P., 2022. Retinal OFF ganglion cells allow detection of quantal shadows at starlight. *Current Biology* 32, 2848-2857.e6. <https://doi.org/10.1016/j.cub.2022.04.092>

REVIEWERS' COMMENTS

Reviewer #1 (Remarks to the Author):

The authors have addressed the raised concerns. One suggestion and one comment accrue from the redlined text:

P1 #18 [suggestion] This sentence seems rather grandiose, and lacks a proper subject. Suggest:

P1 #18 / Here, we resolve this / Here, we address this question /

P4 #18 [Comment] Hahn et al (2023) claim homolog of mouse *_Transient_* alpha cells to parasol cells, and homolog mouse *_Sustained_* alpha cells to beta cells; this nicety could be noted.

Reviewer #2 (Remarks to the Author):

The authors did an outstanding job of supplementing the studies to address my concerns and improve the quality of the manuscript. There are a few minor issues that the authors need to address, before it is ready for publication.

1. The Figshare database url on Page 15 line 19 is either broken or incorrect. Please correct.
2. Page 15, line 5: The sentence should be 'Reported values are mean \pm SEM ...', not 'means \pm SEM'.
3. Since the number of subunits is a parameter tested in the robustness analysis in Supplementary Fig 4, the authors should include the default number of subunits in a-d.
4. Page 14 line 19: Change the phrase to "...and finally, the threshold for the post-retinal nonlinearity".

Reviewer #2 (Remarks on code availability):

The codebase was sufficient to reproduce results/figures in the manuscript.

Reviewer #3 (Remarks to the Author):

In their revised manuscript, the authors have satisfactorily addressed all reviewer comments and suggestions. The detailed discussion of how these results relate to Tinsley et al is especially appreciated and will be a valuable contribution to the literature (especially when the authors' forthcoming study using a single-photon source is also published). All positive comments on the manuscript from the original referee report still apply, and all issues have been addressed.

REVIEWERS' COMMENTS

Reviewer #1 (Remarks to the Author):

The authors have addressed the raised concerns. One suggestion and one comment accrue from the redlined text:

P1 #18 [suggestion] This sentence seems rather grandiose, and lacks a proper subject. Suggest:

P1 #18 / Here, we resolve this / Here, we address this question /

Thank you. We have edited this sentence as the reviewer #1 suggests (P1, line 18).

P4 #18 [Comment] Hahn et al (2023) claim homolog of mouse *_Transient_* alpha cells to parasol cells, and homolog mouse *_Sustained_* alpha cells to beta cells; this nicety could be noted.

Thanks for bringing this up. We have now edited the text accordingly (P4, lines 15–22)

Reviewer #2 (Remarks to the Author):

The authors did an outstanding job of supplementing the studies to address my concerns and improve the quality of the manuscript. There are a few minor issues that the authors need to address, before it is ready for publication.

1. The Figshare database url on Page 15 line 19 is either broken or incorrect. Please correct.

The email advice by sent Figshare (December 15, 2022) tells that the link is activated only after the paper is published: “*To ensure your data can be accessed from your article, add the data DOI:<https://doi.org/10.6084/m9.figshare.21696686> to your manuscript. This DOI link will become active when your data are published. We recommend adding the DOI to both your Data Availability Statement and a reference to the dataset.*”

2. Page 15, line 5: The sentence should be ‘Reported values are mean \pm -SEM ...’, not ‘means \pm -SEM’.

Thanks. We fixed this (P15, line 14).

3. Since the number of subunits is a parameter tested in the robustness analysis in Supplementary Fig 4, the authors should include the default number of subunits in a-d.

The reviewer must be referring to Supplementary Fig. 3. We have now added the number of subunits in the Fig. legend: “*The model assumes 6 subunits ($n = 6$) and 2000 rods per subunit.*” (Supplementary Information, P4).

4. Page 14 line 19: Change the phrase to “...and finally, the threshold for the post-retinal nonlinearity”.

This is now fixed (P14, line 28).

We have now changed the sentence “*and finally the threshold for the final nonlinearity*” to “*and finally, the threshold for the post-retinal nonlinearity*”

Reviewer #2 (Remarks on code availability):

The codebase was sufficient to reproduce results/figures in the manuscript.

Great!

Reviewer #3 (Remarks to the Author):

In their revised manuscript, the authors have satisfactorily addressed all reviewer comments and suggestions. The detailed discussion of how these results relate to Tinsley et al is especially appreciated and will be a valuable contribution to the literature (especially when the authors' forthcoming study using a single-photon source is also published). All positive comments on the manuscript from the original referee report still apply, and all issues have been addressed.

Thanks for the encouraging comments. We appreciate the excitement and interest towards the forthcoming study, as well.